# Passive shaping of intra- and intercellular m6A dynamics via mRNA metabolism

**David Dierks[1], Ran Shachar[1], Ronit Nir[1], Miguel Angel Garcia-Campos[1], Anna Uzonyi[1], David Wiener[1,2], Ursula Toth[3], Walter Rossmanith[3], Lior Lasman[1], Boris Slobodin[4], Jacob H Hanna[1], Yaron Antebi[1], Ruth Scherz-Shouval[5], Schraga Schwartz[1]\***

[1]Department of Molecular Genetics, Weizmann Institute of Science, Rehovot, Israel; [2]Center for Environmental Genomics, University of Washington, Seattle, United States; [3]Center for Anatomy & Cell Biology, Medical University of Vienna, Vienna, Austria; [4]Department of Biochemistry, Rappaport Faculty of Medicine, Technion, Haifa, Israel; [5]Department of Biomolecular Sciences, Weizmann Institute of Science, Rehovot, Israel

## eLife Assessment

This study presents a **fundamental** finding on how levels of m6A levels are controlled, invoking a consolidated model where degradation of modified RNAs in the cytoplasm plays a primary role in shaping m6A patterns and dynamics, rather than needing active regulation by m6A erasers and other related processes. The evidence is **compelling** through its use of transcriptome-wide data and mechanistic modeling. Relevant for any reader with an interest in RNA metabolism, this new framework consolidates previous observations and highlights the importance of careful experimentation for evaluation m6A levels.

**\*For correspondence:**
schwartz@weizmann.ac.il

**Competing interest:** The authors declare that no competing interests exist.

**Abstract** m6A is the most widespread mRNA modification and is primarily implicated in controlling mRNA stability. Fundamental questions pertaining to m6A are the extent to which it is dynamically modulated within cells and across stimuli, and the forces underlying such modulation. Prior work has focused on investigating active mechanisms governing m6A levels, such as recruitment of m6A writers or erasers leading to either 'global' or 'site-specific' modulation. Here, we propose that changes in m6A levels across subcellular compartments and biological trajectories may result from passive changes in gene-level mRNA metabolism. To predict the intricate interdependencies between m6A levels, mRNA localization, and mRNA decay, we establish a differential model 'm6ADyn' encompassing mRNA transcription, methylation, export, and m6A-dependent and -independent degradation. We validate the predictions of m6ADyn in the context of intracellular m6A dynamics, where m6ADyn predicts associations between relative mRNA localization and m6A levels, which we experimentally confirm. We further explore m6ADyn predictions pertaining to changes in m6A levels upon controlled perturbations of mRNA metabolism, which we also experimentally confirm. Finally, we demonstrate the relevance of m6ADyn in the context of cellular heat stress response, where genes subjected to altered mRNA product and export also display predictable changes in m6A levels, consistent with m6ADyn predictions. Our findings establish a framework for dissecting m6A dynamics and suggest the role of passive dynamics in shaping m6A levels in mammalian systems.

## Introduction

*N*6-methyladenosine (m6A) is the most abundant internal modification in mammalian mRNA. Initially discovered five decades ago (*Desrosiers et al., 1974*; *Perry and Kelley, 1974*), research into this modification was catalyzed with the advent of transcriptome-wide detection methodologies, namely m6A-seq/m6A-meRIP (*Dominissini et al., 2012*; *Meyer et al., 2012*). These methods rely on the immunoprecipitation of m6A-containing RNA fragments, followed by high-throughput sequencing and facilitate the extent of m6A methylation stochiometry on the m6A-site, -gene (m6A-GI) and sample level (m6A-SI) (*Dierks et al., 2021*). m6A has been implicated in most steps of mRNA fate and the disruption of the m6A methylation machinery has critical manifestations in diverse systems (*Batista et al., 2014*; *Cui et al., 2017*; *Ke et al., 2017*; *Li et al., 2017*; *Liu et al., 2015*; *Roundtree et al., 2017*; *Shi et al., 2017*; *Wang et al., 2014*; *Wang et al., 2015*). Nonetheless, the most consistently observed molecular outcome of m6A is cytoplasmic mRNA degradation (*Du et al., 2016*; *Lasman et al., 2020*; *Lee et al., 2020*; *Li et al., 2018*; *Uzonyi et al., 2023*; *Zaccara and Jaffrey, 2020*), mediated via cytoplasmic 'm6A readers' of the YTH family (*Du et al., 2016*; *Wang et al., 2014*). In mouse embryonic stem cells (mESCs), m6A appears to be the predominant driver of mRNA stability, explaining roughly 30% of the variation in mRNA half-lives (*Dierks et al., 2021*; *Uzonyi et al., 2023*).

A fundamental question pertaining to m6A is the extent to which it is dynamically modulated within cells (i.e. across subcellular compartments) and across biological stimuli (such as stress responses). In both cases, the literature to date supports the notion that, at first approximation, the m6A methylome is fixed, and yet that m6A levels may be subjected to changes across both dimensions. Specifically, m6A profiles were found to be qualitatively similar across highly different cell types, developmental stages, and tissues (*Liu et al., 2020*; *Schwartz et al., 2014*). Nonetheless, changes in m6A levels – often at only a minority of sites – have been reported across diverse systems, ranging from cellular responses to diverse stressors, infection, disease, and during development (*Anders et al., 2018*; *Knuckles et al., 2017*; *Lichinchi et al., 2016*; *Liu et al., 2023*; *Su et al., 2018*; *Zhou et al., 2015*). Similarly, in considering m6A levels across different compartments within cells, studies have consistently found m6A levels across chromatin-associated, nucleoplasmic and cytoplasmic mRNA fractions to be very similar (*Ke et al., 2017*; *Roundtree et al., 2017*). Nonetheless, slightly higher levels within the chromatin-associated fraction in comparison to the nucleoplasmic and cytoplasmic fractions (*Ke et al., 2017*) were observed in one study, contrasting with a more recent study finding reduced m6A levels in chromatin-associated RNA (*Tang et al., 2024*; *Yang et al., 2022*).

m6A deposition is catalyzed via a megadalton 'm6A–writer' complex, in which METTL3 acts as the catalytic component. m6A is to a considerable extent hard-coded, governed by the presence of the m6A consensus motif (DRACH motif) and distance from splice sites (*Garcia-Campos et al., 2019*; *He et al., 2023*; *Luo et al., 2023*; *Uzonyi et al., 2023*; *Yang et al., 2022*). In light of a 'hard-coded' deposition baseline, it is interesting to dissect how dynamics in m6A level across compartments or biological stimuli can be achieved. The vast majority of the literature has focused on 'active' dynamics, wherein changes in m6A levels were proposed to be a consequence of changes in deposition or removal of m6A at specific sites (*Hess et al., 2013*; *Knuckles et al., 2017*; *Su et al., 2018*; *Zhao et al., 2014*; *Zhou et al., 2015*). Yet, in the context of other types of marks, 'passive' dynamics have been established, in which changes in the level of a modification are not caused by direct deposition or removal but instead result from shifts in metabolism affecting the molecules that carry these modifications. For example, 5mC DNA methylation during early development is primarily thought to be regulated passively via cell division, whereby each division results in a twofold dilution of the DNA 5mC content (*Mayer et al., 2000*; *Rougier et al., 1998*). It is thus intriguing to speculate that m6A deposition might, too, be shaped by passive mechanisms. Nonetheless, a complicating factor in the context of m6A (with respect to 5mC or other instances of classic passive dynamics) is that the methylation mark itself shapes the metabolism of the transcript harboring it. Considering the well-characterized role of m6A in triggering mRNA degradation, m6A levels, therefore, have the potential of being governed by what we term 'semi-passive' dynamics, wherein m6A *actively* triggers the decay of transcripts harboring it, thereby *passively* altering m6A levels. Under such a model, m6A levels, mRNA decay, and subcellular RNA localization are intricately coupled.

We previously established an approach, m6A-seq2, allowing to capture m6A level either for a specific site (m6A-site score) or a metric capturing the overall methylation for a whole gene body (m6A gene index, m6A-GI) (*Dierks et al., 2021*). Inspired by several instances in which we observed

gene-level changes in m6A and where such relationships were manifested differently across different subcellular compartments, we explored whether changes in m6A levels can be attributed to semi-passive dynamics. We establish a model capturing key steps in the life cycle of an mRNA. Despite the simplicity of the model and the fact that it does not allow any site-specific active reshaping of m6A levels, it predicts ample modulation of m6A levels, and captures relationships between mRNA localization, mRNA half-lives, and m6A levels that could be confirmed based on experimental data. We propose that m6A dynamics, when present, are attributable in part to semi-passive regulation, wherein changes in mRNA metabolism impact the susceptibility of a transcript to m6A-dependent decay, thereby altering m6A levels.

## Results

Previous studies have explored active mechanisms via which m6A levels could be modulated. We sought to examine whether changes in m6A could be passive consequences of changes in mRNA

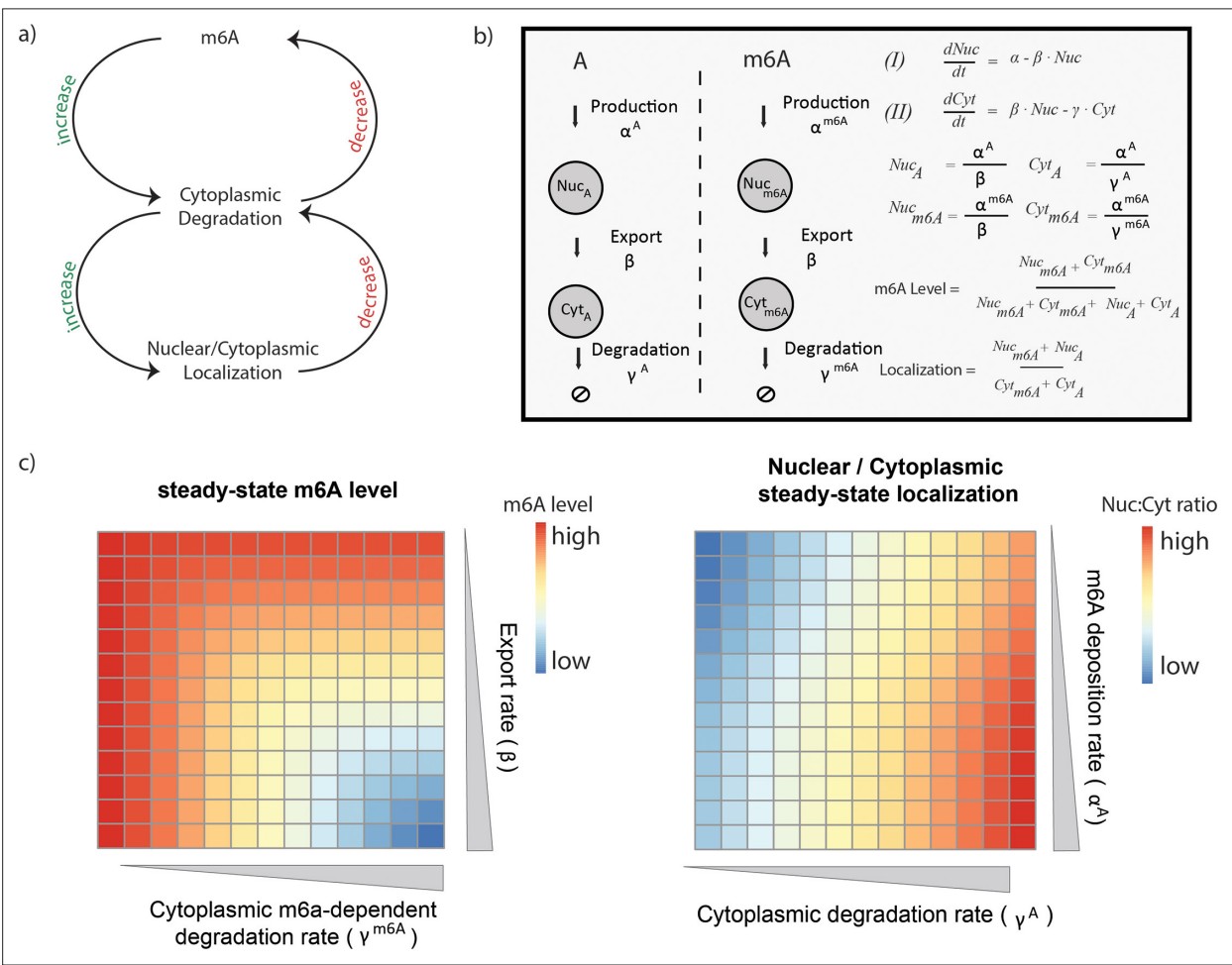

**Figure 1.** m6ADyn simulates subcellular transcript localization and m6A methylation level sensitivity to parameters. (**a**) Schematic representation of the circular relationship between m6A methylation level, nuclear:cytoplasmic localization, and mRNA degradation. (Top) m6A enhances cytoplasmic degradation, while cytoplasmic m6A facilitated degradation decreases m6A level. (Bottom) Cytoplasmic degradation enhances relative nuclear abundance of a gene, while increased nuclear localization reduces susceptibility to m6A-mediated cytoplasmic degradation, thereby increasing stability. (**b**) Depiction of the life cycle of an RNA, as modeled by m6ADyn, along with rates governing the transitions from one step to another (left). Equations for nuclear (Nuc) and cytoplasmic (Cyt) gene abundance based on three rates, production α, export β, and cytoplasmic degradation γ. Solutions, under steady-state conditions, for nuclear and cytoplasmic abundances of methylated and unmethylated transcripts, for overall methylation levels, and for relative nuclear:cytoplasmic levels. (**c**) Heatmap indicating the simulated steady-state m6A level (left) and steady-state Nuc:Cyt localization (right) for a specific set of parameters. In both cases a gene was simulated using 1 for all the rates other than the rates indicated in columns or rows, for which we used increasing values (0.08–3.5).

metabolism. The motivation for exploring this model was the realization that m6A levels both *reflect* and *drive* mRNA intracellular metabolism. Specifically, we considered a simple model wherein (1) m6A promotes mRNA decay, and (2) such decay occurs exclusively in the cytoplasm. This model intuitively links m6A levels and mRNA localization (nucleus/cytoplasm) (*Figure 1a*). On the one hand, under this model, changes in m6A *cause* changes in intracellular mRNA distribution: An increase in cytosolic m6A level of a given gene will trigger selective decay of cytoplasmic m6A-harboring transcripts *causing* increased nuclear:cytoplasmic ratios of the gene. On the other hand, under this model, changes in mRNA localization will lead to changes in m6A levels: An increase in relative nuclear abundance of an mRNA leads to decreased cytoplasmic m6A-dependent degradation, thereby increasing overall m6A levels (*Figure 1a*). Thus, under these model assumptions, m6A levels, cytoplasmic decay, and subcellular localization are deeply intertwined, with changes in m6A driving and reflecting mRNA metabolism.

To model the complex interplay between m6A levels, mRNA localization, and mRNA stability, we established 'm6ADyn', a differential-equation-based model. In m6ADyn, a transcript is produced in the nucleus at a rate of α, gets exported from the nucleus into the cytoplasm at a rate of β, where it is degraded at the rate of γ. To capture the difference in the metabolism of m6A- and non-containing mRNA, we established two sets of parameters: $\alpha^A$, $\beta^A$, and $\gamma^A$ representing production, export, and decay of non-methylated transcripts, respectively, and $\alpha^{m6A}$, $\beta^{m6A}$, and $\gamma^{m6A}$ representing these rates for methylated counterparts. Throughout our simulations, unless indicated otherwise, we assumed (1) that the export rates of methylated and unmethylated transcripts were identical ($\beta^A = \beta^{m6A}$), and we will therefore refer to these rates as β, and (2) that the fate of methylated mRNA differed from that of its unmethylated counterparts exclusively in increased susceptibility to cytoplasmic decay, which was implemented by multiplying the unmethylated decay rate ($\gamma^A$) by a constant ($\gamma^{m6A} = S * \gamma^A$, with $S > 1$). Under steady-state conditions (e.g. under normal growth conditions), the model equations can be solved analytically to determine mRNA and m6A levels in the nucleus and in the cytoplasm (*Figure 1b*).

m6ADyn further enables us to determine the full dynamical response of mRNA and m6A levels under non-steady-state conditions. For instance, m6ADyn allows us to explore how the modulation of parameters governing export rate or cytoplasmic m6A-dependent decay *impacts* m6A levels. Increasing either m6A-dependent cytoplasmic degradation ($\gamma^{m6A}$) or export (β) leads to reduced m6A levels – the former due to depletion of m6A-harboring transcripts, and the latter due to increased levels of m6A-harboring mRNAs in the cytoplasm, rendering them susceptible to m6A-dependent decay (*Figure 1c*, left). In parallel, m6ADyn also allows evaluation of how different aspects of metabolism are *impacted* by m6A levels. For example, increasing the m6A deposition rate ($\alpha^{m6A}$) leads to increased relative nuclear localization, and given that nuclear m6A-harboring transcripts are not subject to selective decay, this leads to an increase in overall m6A levels (*Figure 1c*, right). Thus, m6ADyn allows us to determine how m6A levels drive and reflect mRNA metabolism.

We note that the assumptions made by m6ADyn are reductionist and possibly over-simplistic, given that m6A has been tied, among others, to production, transcription termination, nuclear decay, and export (*Hsu et al., 2019*; *Ke et al., 2017*; *Knuckles et al., 2017*; *Roundtree et al., 2017*; *Slobodin et al., 2020*; *Yang et al., 2019*), all of which are either not parameterized by m6ADyn or not assumed to be different between m6A- and non-harboring transcripts (of note: we do expand m6ADyn, below, to test a subset of these). Yet, despite its simplicity, the predictions of this framework are in many cases non-intuitive, and modeling them provides an opportunity to explore which aspects of m6A dynamics can or cannot be explained using only the above-described simple and well-supported assumptions.

## Testing m6ADyn predictions under steady-state conditions

We sought to assess the extent to which relationships predicted by m6ADyn under steady-state conditions are supported by experimental data. We began with two predictions on how mRNA metabolism impacts m6A levels. A first prediction of m6ADyn concerns heterogeneity in m6A levels across subcellular fractions: m6ADyn predicts that *nuclear m6A levels* per gene would be invariably higher than the corresponding *cytoplasmic levels*, given the selective removal of m6A-harboring transcripts from the cytoplasm ($\gamma^{m6A} = S * \gamma^A$, with $S > 1$). In m6ADyn, nuclear and cytoplasmic m6A can be analytically calculated for each gene in each compartment, defined as the fraction of transcripts harboring m6A divided by the overall number of transcripts in that compartment (Materials and methods, *Equations*

*2 and 3*). To confirm this experimentally, we fractionated NIH3T3 cells into nuclear and cytoplasmic fractions in duplicates and applied m6A-seq2 to all samples, following which we calculated m6A levels at the sample level and gene levels (m6A-SI and m6A-GI) across both compartments. Consistent with m6ADyn predictions, we found that m6A levels were higher in the nuclear fraction both at the sample and at the gene level, with the vast majority of genes (~82%) exhibiting higher methylation levels in the nucleus (*Figure 2—figure supplement 1a* and *Figure 2a*). These trends were also recapitulated in an independent experiment performed in primary mouse embryonic fibroblasts (MEFs) (*Figure 2— figure supplement 1b*).

A second prediction of m6ADyn concerns heterogeneity in m6A levels between genes: m6ADyn predicts that *m6A levels* across the entire cell are higher in genes that are relatively more abundant in the nucleus, given that they are least susceptible to m6A-dependent decay (*Figure 2b*, left). To assess whether this could be experimentally confirmed, we performed RNA-seq on nuclear and cytoplasmic fractions derived from WT and METTL3KO mESCs, which allowed the transcriptome-wide estimation of relative nuclear:cytoplasmic localization. Correlating the WT fractionation data against a previously acquired dataset capturing m6A levels per gene in mESCs (*Dierks et al., 2021*) revealed that indeed m6A levels were higher in more nuclear localized genes, consistent with m6ADyn predictions (*Figure 2b*, right). We made similar observations also in NIH3T3 and in HEK293T cells for which we acquired fractionation data, confirming the robustness of these results (*Figure 2—figure supplement 1c, f*). To control for potential technical covariates in antibody-based mapping, we additionally used previously published dataset of 40,109 m6A sites, captured at single-nucleotide resolution, identified via GLORI, an antibody independent method relying on resistance of m6A sites to chemical deamination (*Liu et al., 2023*). We aggregated the signal stemming from individual m6A sites within a gene into a GLORI-gene index (GLORI-GI) (Materials and methods). A comparison of GLORI-GIs with our m6A-seq-derived m6A-GIs showed a high correlation of *R* = 0.71 (*Figure 2—figure supplement 1d*), supporting the robustness of m6A-GIs. Moreover, GLORI-GIs were higher in more nuclear genes (*Figure 2—figure supplement 1e*), consistent with our observations on the basis of m6A-GIs.

We next sought to assess m6ADyn predictions relating to how changes in m6A levels affect mRNA localization and stability. We first focused on mRNA subcellular localization. m6ADyn predicts that loss of methylation (as in Mettl3 KO conditions) in genes that are highly methylated in wild-type conditions would lead to increased cytoplasmic presence (and hence reduced nuclear:cytoplasmic levels), whereas such an effect would be less pronounced in poorly methylated genes. Consistently with this model prediction, a comparison of localization in WT and m6A-depleted mESCs revealed that loss of m6A results in an increased propensity for highly methylated genes to become cytoplasmic (*Figure 2c*). Notably, relationships between m6A levels and localization have previously served as potential indications of m6A impacting nuclear export (*Edens et al., 2019*; *Hsu et al., 2019*; *Kim et al., 2021*; *Lesbirel et al., 2018*; *Roundtree et al., 2017*). The results demonstrate that even when assuming identical export rates between m6A- and non-containing transcripts, substantial differences in m6A gene level can be explained solely by differences in subcellular localization and finally through cytoplasmic decay (*Figure 2b, c*).

A final set of predictions of m6ADyn concerns m6A-dependent decay. m6ADyn predicts that (1) cytoplasmic genes will be more susceptible to increased m6A-mediated decay and (2) more methylated genes will undergo increased decay (*Figure 2d*, left). To explore these predictions, we relied – in addition to the above-described fractionation and m6A-seq2 measurements – also on previously published Actinomycin D (ActD) based mRNA stability measurements in WT and Mettl3 KO mESCs (*Ke et al., 2017*). For both experimental data, and model simulations, mRNA half-lives were acquired via linear modeling of the logarithmic decrease in mRNA levels over the time course following transcriptional inhibition (Materials and methods). Strikingly, the experimental data supported the dual, impact of m6A levels and localization on mRNA stability (*Figure 2d*, right). This trend could also be recapitulated in HEK293T, when using available m6A-seq, subcellular fractionation, and mRNA stability measurements following treatment HEK293T cells with either STM2457 (a small molecule inhibitor of METTL3) or mock (DMSO) (*Figure 2—figure supplement 1g*; *Yankova et al., 2021*). In our previous studies, we were able to attribute roughly 25% of the variability in mRNA half-lives to m6A levels (*Dierks et al., 2021*; *Uzonyi et al., 2023*). Here, however, analyzing this relationship through the lens of subcellular localization revealed that among cytoplasmic genes – most susceptible to m6A-dependent decay – the association between localization and m6A is substantially stronger,

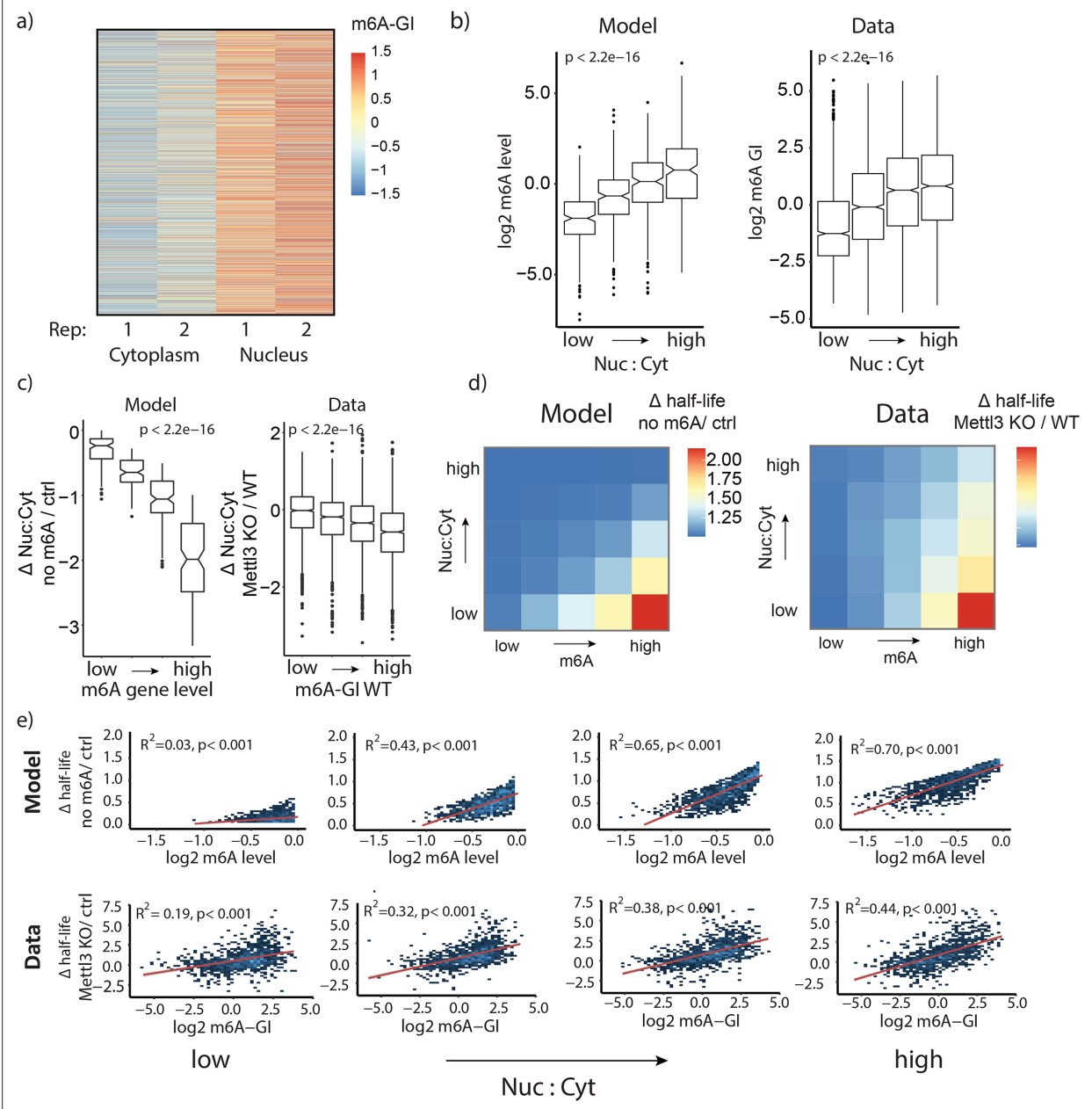

**Figure 2.** Predictive modeling unravels interactions of steady-state m6A level, mRNA localization, and stability. (**a**) m6A-GIs of a nuclear fraction and a cytoplasmic fraction, based on m6A-seq2 measurements in NIH3T3 cells in two replicates ($n = 6935$ genes). (**b**) Impact of nuclear:cytoplasmic localization on m6A levels. (Left) m6ADyn predicted m6A levels, plotted as a function of m6ADyn predicted nuclear:cytoplasmic ratios, for simulated genes ($n = 2000$). Parameters for alpha, beta, and gamma were sampled independently from a gamma distribution (rate = 1, shape = 1). (Right) Experimentally measured m6A gene levels (m6A-GIs, $y$-axis) as a function of relative nuclear:cytoplasmic steady-state localization measured in mouse embryonic stem cell (mESC) WT, divided into four equally sized bins (right) ($n = 8858$ genes). (**c**) Distribution of difference in nuclear:cytoplasmic localization as a function of m6A levels for modeled and experimental data, binned into four equally sized bins. (Left) Simulations were conducted for $n = 2000$ genes. m6A-depleted conditions were modeled using $\gamma^{m6A} = \gamma^A$, that is no m6A facilitated cytoplasmatic degradation, while in the control conditions, we used $\gamma^{m6A} = 10 * \gamma^A$. (Right) The difference in nuclear:cytoplasmic localization measured in WT versus METTL3 KO mESCs, plotted as a function of m6A-GIs measured in mESCs WT ($n = 4527$ genes). (**d**) Impact of m6A levels and nuclear:cytoplasmic localization on mRNA half-life. (Left) Heatmap showing simulated m6A-dependent half-lives, binned by simulated steady-state Nuc:Cyt ratios and m6A-gene level based on 2000 simulated genes using m6ADyn (rate sampling performed as in a). Genes were binned into 25 bins based on their m6ADyn m6A levels ($x$-axis) and their m6ADyn nucleus:cytoplasmic levels ($y$-axis). The median simulated m6A-dependent half-life per gene is color-coded. (Right) Corresponding values for experimentally derived data, on the basis of m6A-dependent half-lives (mESC Mettl3 KO half-life/mESC WT half-life), nuclear localization, and m6A data generated in WT mESCs for a total of 5354 for which we had mESC WT Nuc:Cyt ratios, Mettl3 KO and WT half-lives, and m6A-GI values. (**e**) Associations between METTL3-dependent

*Figure 2 continued on next page*

*Figure 2 continued*

half-lives and m6A levels as a function of nuclear:cytoplasmic localization, binned into four equally sized groups. (Top) Scatterplots of m6ADyn-derived half-lives as a function of m6ADyn-derived m6A levels ($n$ = 2000 gene simulations, rates sampled as in a). (Bottom) METTL3-dependent half-lives (Mettl3 KO/WT HL) as a function of m6A-GIs based on measurements in mESCs ($x$-axis).

The online version of this article includes the following figure supplement(s) for figure 2:

**Figure supplement 1.** Steady-state predictions of m6ADyn across cell lines.

**Figure supplement 2.** Comparison of m6ADyn predictions based on different assumptions.

with 44% of the variability in METTL3-dependent decay being attributable to m6A levels. Conversely, among nuclear genes this relationship is substantially weaker ($R^2$ = 0.19). These trends are qualitatively predicted by m6ADyn (*Figure 2e*) and were further recapitulated using HEK293T measurements (*Figure 2—figure supplement 1h*).

We next sought to assess whether alternative models could readily predict the positive correlation between m6A and nuclear localization and the negative correlations between m6A and mRNA stability. First, we assessed how nuclear decay might impact these associations by introducing nuclear decay as an additional rate, $\delta$. We found that both associations were robust to this additional rate (*Figure 2—figure supplement 2a–c*). Next, we sought to assess how m6A-facilitated export ($\beta^{m6A}$ = 10 * $\beta^A$) might impact the predictions by the model. Given the selective depletion of m6A from the nucleus, due to the facilitated export of the methylated transcript from the nucleus, the associations between m6A and nuclear localization were reversed (*Figure 2—figure supplement 1a–c*). This model is, therefore, inconsistent with measurements. Finally, we also assessed an inverse model wherein m6A is inhibitive of export ($\beta^{m6A}$ = 0.1 * $\beta^A$). This model recovers the positive association between m6A and nuclear localization but gives rise to a positive association between a gene's m6A level and its stability – again contrasting with measurements (*Figure 2—figure supplement 1a–c*). Thus, among the tested models, only ones in which m6A negatively impacts stability allow qualitatively recovering both the association with localization and with stability.

## Testing m6ADyn predictions upon perturbations of mRNA metabolism

Next, we sought to assess m6ADyn predictions for how perturbations of mRNA metabolism impact m6A levels. m6ADyn predicts that decreasing production values (i.e. decreasing $\alpha$ rates) results in a decrease in m6A levels. Moreover, m6ADyn predicts that the rate at which m6A levels decrease is inversely correlated with mRNA stability, that is that m6A levels will decrease faster in more short-lived genes (*Figure 3a*). To test this prediction, we employed m6A-seq2 to monitor m6A levels along a 6-hr time course following the treatment of mESC cells with ActD. Consistent with the m6ADyn predictions (*Figure 3b*, left), gene-level measurements of m6A displayed a subtle yet consistent decrease in m6A levels (*Figure 3b*, right). This drop was also observed at the sample level (*Figure 3—figure supplement 1a*), consistent also with previously reported mass-spec-based measurements that showed a decrease in m6A/A ratio following ActD treatment (*Roundtree et al., 2017*). Similarly, treatment of MCF7 cells with Camptothecin (CPT), a topoisomerase inhibitor that stalls RNAPII elongation, thereby reducing production rates (*Listerman et al., 2006*), led to reduced m6A levels, in this case with much more dramatic effect sizes on both the sample and gene levels (*Figure 3—figure supplement 1b*, *Figure 3c*). As predicted by m6ADyn (*Figure 3d*, left), the decrease in m6A was stronger in genes with lower half-lives (*Figure 3d*, right). Consistently, following both ActD and CPT administration, we observed that the decrease in m6A levels was inversely proportional to the stability of the genes (*Figure 3—figure supplement 1c*, *Figure 3d*).

To further test predictions of m6ADyn, we next sought to perturb m6A-dependent decay to assess whether predictions of m6ADyn following perturbations of $\gamma^{m6A}$ could be experimentally confirmed. m6A-Dyn predicts that elimination of m6A-dependent decay (e.g. $\gamma^{m6A}$ = $\gamma^A$) will result (1) in increased overall methylation levels (*Figure 3e*) and (2) in decreased nuclear/cytoplasm localization due to stabilization of cytoplasmic transcripts. Moreover, this impact on stabilization would be predicted to be stronger for the more methylated genes. Given that m6A-mediated decay is primarily thought to be mediated via the three cytoplasmic YTHDF proteins (YTHDF1, YTHDF2, YTHDF3) (*Du et al., 2016*; *Lasman et al., 2020*; *Zaccara and Jaffrey, 2020*), we next conducted a triple knockdown of the YTHDF proteins in NIH3T3 cells followed by whole-cell m6A-seq2 measurements and subcellular

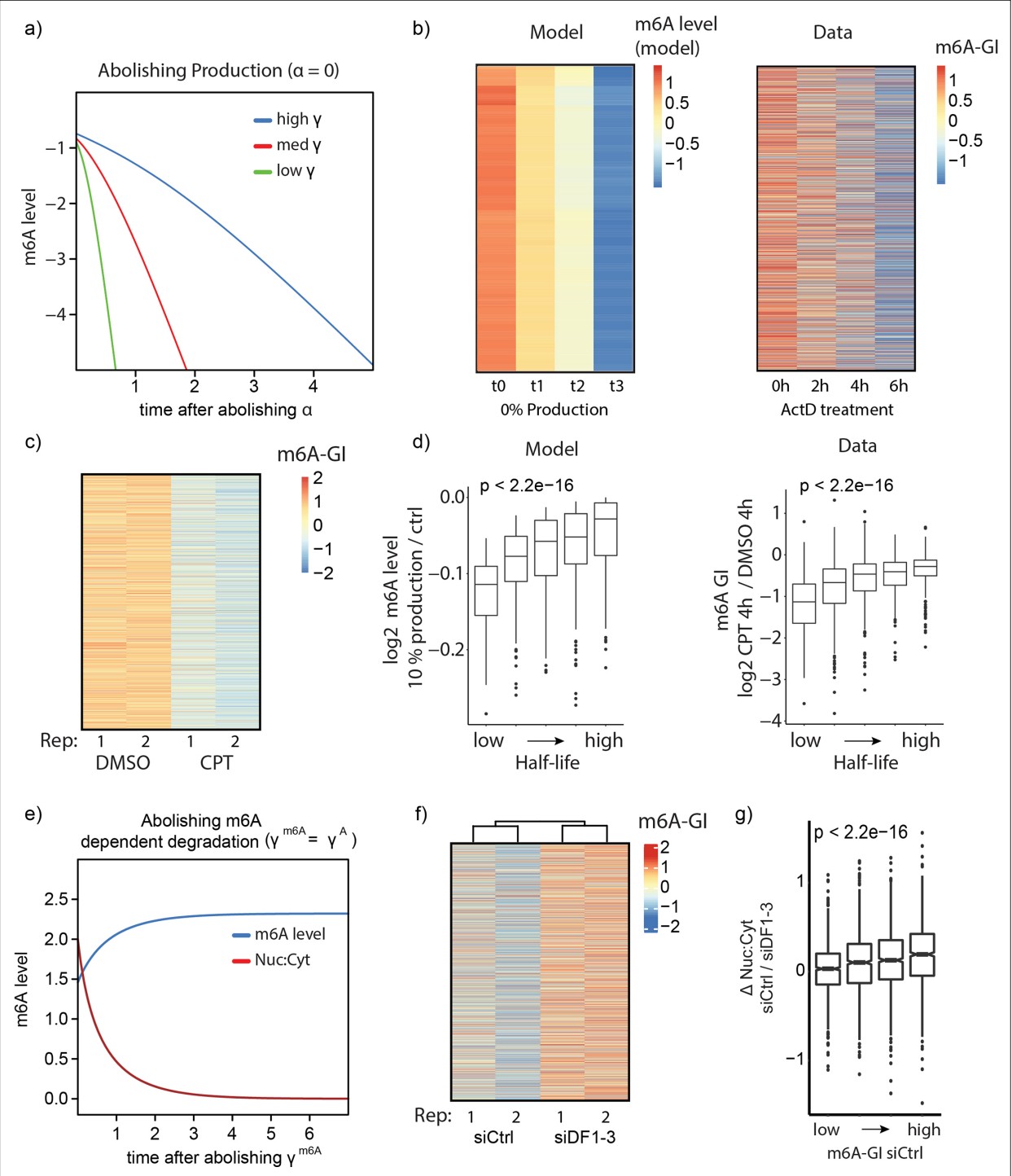

**Figure 3.** Perturbations of RNA production or m6A methylation lead to predicted outcomes in localization and methylation level. (**a**) Simulation of normalized m6A gene level after abolishing the production rate ($\alpha = 0$) for three simulated genes with different cytoplsamatic degradation rates. The three genes were simulated with identical rates (all rates equal 1, except $\gamma^A$ and $\gamma^{m6A}$). The ony difference were three increasing degradation rates ($\gamma^A$, with $\gamma^{m6A} = 10 * \gamma^A$) and are indicated. (**b**) Comparison of model derived to experimentally measured changes in m6A levels following transcriptional inhibition. (Left) m6A levels as predicted by m6ADyn on the basis of 2000 sampled genes for timepoints following perturbations in which production rate ($\alpha$) is set to zero (rates were sampled as in **Figure 2a**). (Right) Experimentally (m6A-seq2) measured m6A-GIs in an A9 mouse fibroblast cell line along the indicated timepoints following ActD-mediated transcriptional inhibition. In both cases, the heatmaps are scaled across rows ($n = 8788$). (**c**) Heatmap of row-scaled m6A-GIs of MCF7 cells treated with Camptothecin (CPT) (4 hr) and the control group DMSO (4 hr) ($n = 6210$). (**d**) Changes in m6A levels upon reducing production as a function of half-life. (Left) m6ADyn simulated values ($n = 1000$). WT rates were sampled as in **Figure 2a**. Reduced production

*Figure 3 continued on next page*

*Figure 3 continued*

was simulated by setting the production rate (α) to 10% of its initial level. (Right) Distribution of m6A-GI changes following CPT treatment. Genes were binned into five equally sized bins based on their half-lives (right), human mRNA half-lives were obtained from *Agarwal and Kelley, 2022* (*n* = 6059). (**e**) Simulated changes over time of m6A levels (blue) and nuclear:cytoplasmic (red) localization upon elimination of m6A-dependent degradation, by setting $\gamma^{m6A}$ to equal $\gamma^A$. (**f**) Row-scaled m6A-GIs based on m6A-seq2 measurements of NIH3T3 cells treated with siCtrl or with siRNAs targeting all three Ythdf1, -2, -3 (siDF1–3) (*n* = 5915). (**g**) Distribution of the fold-change of Nuc:Cyt ratio of the siCtrl compared to siDF1–3, with the genes binned by the steady-state m6A level (m6A-GI in the siCtrl sample) (*n* = 5915).

The online version of this article includes the following figure supplement(s) for figure 3:

**Figure supplement 1.** m6A methylation changes upon mRNA metabolism perturbations.

**Figure supplement 2.** Outliers in m6ADyn predictions following mRNA metabolism perturbations.

fractionation coupled with RNA-sequencing. While the knockdown substantially reduced RNA levels of YTHDF1 and YTHDF2, it did not reduce YTHDF3 levels (*Figure 3—figure supplement 1d*). Nonetheless, there was an overall increase, albeit subtle, in m6A levels on the gene and sample levels (*Figure 3—figure supplement 1e*, *Figure 3f*). Moreover, as anticipated by m6ADyn, we noted a reduction in the nuclear/cytoplasmic levels of more highly methylated genes following this perturbation (*Figure 3g*). These analyses thus support the relevance of m6ADyn for interpreting dynamic changes in m6A levels following alterations in mRNA metabolism. Nonetheless, the relatively subtle observed effect sizes hint at limitations of the model and/or at pleiotropic effects induced by the metabolic perturbations (see Discussion).

## Changes in m6A levels in the biological response to heat stress are secondary to changes in mRNA metabolism

Finally, we tested whether our model can serve as a framework for exploring how altered mRNA metabolism during a biological response leads to alterations in m6A levels. The cellular response to heat shock is well documented to be associated with significant alterations in mRNA metabolism, characterized by a pronounced decrease in global export and an increase in the production of heat stress response (HSR) genes (*Mahat et al., 2016*; *Shalgi et al., 2014*; *Trinklein et al., 2004a*; *Trinklein et al., 2004b*). Additionally, the cellular heat shock response has been implicated in differential m6A methylation of heat shock response genes (HSR-gene) at hundreds of m6A sites (*Knuckles et al., 2017*; *Liu et al., 2023*; *Yang et al., 2022*; *Meyer et al., 2015*; *Zhou et al., 2015*). Both the increase in mRNA production rates and the reduction in export are predicted by m6ADyn to result in increasing m6A levels, thereby rendering heat shock an attractive model for testing to what extent observed changes in m6A levels could be directly explained by changes in mRNA metabolism (*Figure 4a*).

We first sought to confirm previous findings pertaining to the heat-shock-induced export block. We applied a single-molecule FISH protocol using a poly(dT) probe directed against poly(A) mRNA in MEFs. We assessed global changes in mRNA localization in cells grown under normal conditions (non-heat shock, NHS) at 37°C, cells subjected to 1.5 hr of heat shock at 43°C (heat shock, HS) and cells in recovery at 37°C for 1 and 4 hr, following the stress. We observed a dramatic increase in the nuclear:cytoplasmic ratio of mRNA upon heat stress treatment (*Figure 4b, c*), consistent with previously documented nuclear retention of mRNA upon heat stress (*Shalgi et al., 2014*).

We next applied m6A-seq2 to a heat stress and recovery time course. The experiment was performed in primary MEF cells, in biological triplicates, using the same timepoints as in the FISH measurements with the addition of an early timepoint 45 min following heat shock (*Figure 4d*). We obtained m6A-GIs for ~8100 genes expressed at sufficient levels across all samples. In general, m6A levels across genes were highly correlated between all timepoints, consistent with previous studies demonstrating that m6A is, to a considerable extent, a constitutive feature of the mRNA. Nonetheless, subtle quantitative differences were apparent (*Figure 4—figure supplement 1a*). To capture the most dominant trends in m6A levels along this time course, we performed a principal components analysis (PCA) on the m6A-GIs of all samples in the time course. The first principle component (PC1) captured a continuous increase (or decrease) of m6A methylation along the heat stress time course (*Figure 4e*). We hence use PC1 below as an unbiased metric for ranking genes based on whether they undergo induced m6A levels over time (high PC1) or reduced (low PC1). While this manuscript focuses primarily on m6A levels per gene, we confirmed the change of m6A methylation during the heat shock

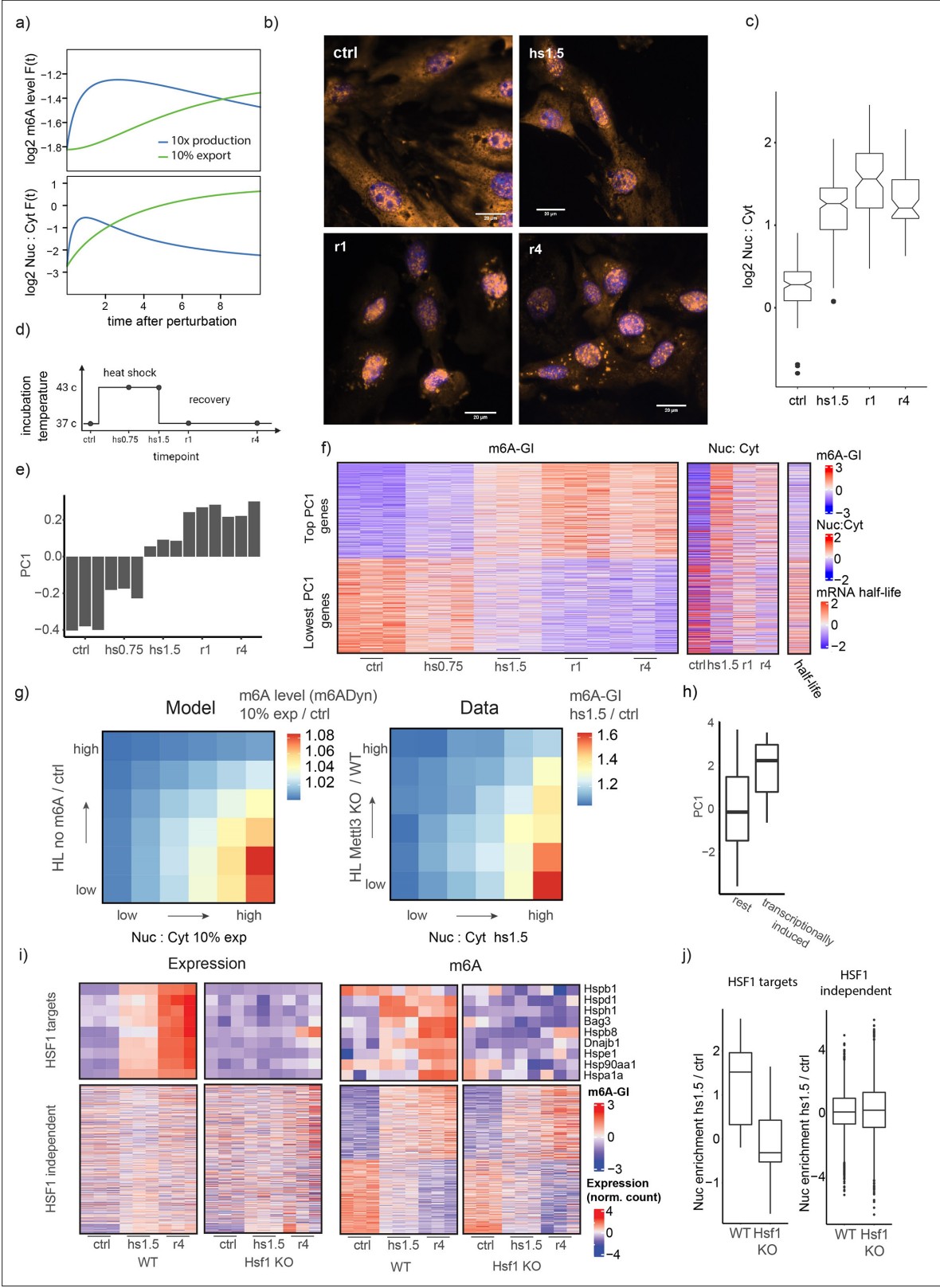

**Figure 4.** Changes in m6A gene modification under heat shock conditions are reflected by changes in mRNA metabolism. (**a**) m6ADyn prediction of m6A gene level (top) and the genes relative localization Nuc:Cyt ratio (bottom) modeled by m6ADyn over time after perturbation of the export or production rate. The perturbations are the increase of the production rate by a factor of 10 or the decrease of the export rate to 10%. (**b**) Representative FISH poly-A staining signal and DAPI signal merged for timepoints of the cellular heat shock time course experiment (scale bar = 20 μm). (**c**)

*Figure 4 continued on next page*

*Figure 4 continued*

Quantification of the poly-dt staining signal comparing mean intensity for nucleus/mean cytoplasmic signal. (**d**) Scheme of a heat shock time course experiment timepoints: ctrl/NHS, heat stress at 43°C 0.75 or 1.5 hr and two timepoints after the recovery at 37°C at 1 hr (r1h) and at 4 hr (r4h). (**e**) PC1 of a principal components analysis (PCA) of the m6A-GIs based on the m6A-seq2 experiment. (**f**) Heatmap depicting m6A-GIs, nuclear:cytoplasmic localization data, and mRNA stability for 1000 genes with the highest and lowest PC1 loading (plotted from high to low). (Left) Depiction of row-scaled m6A-GIs over the indicated timepoints and replicates. (Middle) Row-scaled Nuc:Cyt ratios. (Right) Available half-life measurements of NIH3T3 cells (column scaled) (*Agarwal and Kelley, 2022*). (**g**) Fold-change in m6A level following an export block, as a function of difference in nuclear:cytoplasmic localization and METTL3-dependent half-lives. Values are plotted in the form of a heatmap, binned into 6 bins along each dimension (yielding a total of 36 bins). (Left) m6ADyn simulations (*n* = 1000, rates sampled as in *Figure 2a*) comparing steady-state m6A level and m6A level after reduction of export rates, with beta reduced to 10% of its WT levels. (Right) Experimentally measured fold-changes of m6A level of heat shock 1.5 hr in comparison to steady-state control level, depicted as a function of changes in nuclear:localization and METTL3-dependent half-lives (half lives in WT compared to METTL3 KO cells). (**h**) Boxplot showing the distribution of PC1 genes of genes binned by PRO-seq data (*Mahat et al., 2016*). PRO-seq data were divided into two bins: genes displaying a 10-fold increase in PRO-seq coverage upon heat shock (transcriptionally induced) and all remaining genes (rest). (**i**) Heatmap depicting row-scaled normalized expression level (left) and m6A-GIs (right) based on three timepoints (triplicates) of Hsf1-KO cells and the five timepoints of the WT cells. (Top) Nine HSF1-target HSR-genes and (bottom) 500 highest and lowest PC1 (based only on the WT-sampled) genes. (**j**) Distribution of relative Nuc:Cyt localization of HSF1-targets (left) and for the other HSF1-independent genes (right). Based on Nuc:Cyt fractionation experiment based on WT and Hsf1 KO mouse embryonic fibroblasts (MEFs) one-sided Mann–Whitney *U* test p values are indicated, testing for a higher Nuc:Cyt values in the wild-type fractionation sample.

The online version of this article includes the following figure supplement(s) for figure 4:

**Figure supplement 1.** m6A methylation and nuclear localization dynamics during heat shock response.

time course with SCARLET for three chosen m6A sites in genes that showed a dynamic increase in m6A upon heat stress (high PC1 genes) (*Figure 4—figure supplement 1b*).

We next sought to understand whether – consistently with m6ADyn predictions – the changes in m6A levels could be linked with differences in mRNA subcellular localization during heat stress. We monitored relative subcellular localization over the course of heat stress via fractionation of cells into nuclear and cytosolic fractions, followed by RNA sequencing. Remarkably, we observed that the genes undergoing increased methylation upon heat stress (high PC1 genes) became dramatically more nuclear during heat stress (*Figure 4f*). Conversely, the genes with decreased methylation (low PC1 genes) exhibited a reversed trend (*Figure 4f*). Overall, PC1 loading of genes and nuclear:cytoplasmic fractionation display a correlation of *R* = 0.4 (*Figure 4—figure supplement 1c*). To exclude m6A as a driver of these mRNA metabolic changes, we performed an siRNA-mediated knockdown of *Wtap*, a component of the m6A–'writer' complex and shown to be an effective target for transcriptome-wide m6A depletion. si*Wtap* treated cells were subjected to a heat shock time course experiment followed by transcriptional profiling in nuclear and cytoplasmic compartments. We found that genes with high PC1 values in the WT experiment showed a similar trend of increased nuclear abundance upon heat stress also in the si*Wtap* treated cells, suggesting that m6A is unlikely to be the driver of these changes (*Figure 4—figure supplement 1e*). These results thus tie alterations in m6A levels to alterations of subcellular localization induced during heat stress and are consistent with the notion that disrupted export during heat stress could underlie the systematic increase in m6A levels observed in heat stress.

In our analysis of the dataset and attempting to understand why only certain genes exhibit induced m6A levels than others, we made two observations: First, high PC1 genes (displaying induced m6A levels during heat stress) tend to have substantially higher turn-over rates (shorter half-lives) in comparison to low PC1 genes. Second, we observed that high PC1 genes become more nuclear upon heat stress (*Figure 4f*). To assess whether differences in nuclear localization upon export block, combined with differences in m6a-dependent half-lives between genes were sufficient to account for observed differences in m6A levels under heat stress conditions, we used m6ADyn to predict changes in methylation levels following a global reduction of export (down to 10% of steady-state export rates). We observed that METTL3-dependent half-life and Nuc:Cyt ratio of genes following export block were conditionally independent of each other, with each of them individually (and both of them additively) negatively correlating with extent of m6A induction (*Figure 4g*, left). Remarkably, highly similar trends were observed when analyzing the experimental dataset in a similar manner, binning genes based on their METTL3-dependent stability and based on their relative Nuc:Cyt localization after 90 min heat stress treatment (*Figure 4g*, right). This analysis thus highlights how the complex interaction between diverse aspects of RNA metabolism and m6A levels can be captured by m6ADyn.

A closer inspection of high PC1 genes (in which m6A is induced during the cellular stress response) revealed that these were enriched in 'cellular stress response' gene ontology, comprising classical heat stress response (HSR-) genes such as Hspa1a, Hspb1, Hsp1d, Hsp1e, Hsph1, and Dnajb1 (*Figure 4—figure supplement 1e*). An analysis of Pro-seq data, which evaluates RNA Pol II occupancy as a proxy for transcriptional rate in MEFs during heat stress, confirmed that genes that showed the highest transcriptional induction are genes that have a high PC1 loading, that is show increased m6A level upon heat stress (*Figure 4h*). This was of interest to us, as a prediction of m6ADyn is that increased production rates will result in increased m6A levels (*Figure 4a*). We, therefore, sought to validate that the increased methylation observed in these genes was indeed a consequence of their transcriptional induction. We reasoned that perturbing heat shock factor 1 (HSF1), the central transcription factor governing the heat stress response, could allow assessing whether the increased m6A levels in HSF1-transcriptionally induced targets are truly attributed to increased production or potentially due to some other mode of regulation. To explore this, we applied m6A-seq2 to a heat stress time course in HSF1-KO primary MEFs. We first confirmed that also within this dataset, HSF1-independent genes displayed similar m6A methylation dynamics as observed in the WT cells (*Figure 4i*, bottom). In stark contrast, while HSF1-target genes remained expressed at basal levels also within HSF1-KO cells, their m6A dynamics were completely lost (*Figure 4i*, top). Consistently, fractionation experiments in HSF1-KO cells revealed that HSF1-target genes were no longer enriched in the nucleus, in stark contrast to the HSF-independent counterparts. Furthermore, fractionation experiments of heat stress treated HSF1-KO cells followed by RNA-seq confirmed that the HSF1-target genes do not experience an increase in nuclear enrichment during heat stress, unlike in the wild-type cells. Conversely, HSF1-independent genes demonstrate a similar increase in nuclear enrichment in both HSF1-KO and wild-type cells (*Figure 4j*). These analyses thus demonstrate that the m6A increase observed within HSF1-target genes is a consequence of the transcriptional induction, in line with m6ADyn predictions that induced production should be sufficient to lead to increased m6A levels.

To summarize, our results indicate that during the heat stress response, m6A accumulates within a large set of genes, primarily within targets that undergo increased nuclear localization. This is likely primarily due to a block in export, but – in a subset of genes – due to increased transcription. In both cases, the impact on nuclear:cytoplasmic levels and on m6A levels are captured well by the m6ADyn framework. These results thus highlight the utility of the m6ADyn framework in capturing and predicting m6A dynamics in the context of a complex cellular response.

## Discussion

Until recently, the rules governing the selectivity of m6A deposition were shrouded in mystery. In the absence of an understanding of how m6A deposition is governed, we also lacked a framework for considering rules governing m6A dynamics. Consequently, diverse models were proposed for both aspects, typically relying on some form of 'active' recruitment of the methyltransferase complex and/or of m6A erasers to target sites (*Anders et al., 2018*; *Huang et al., 2019*; *Knuckles et al., 2018*; *Knuckles et al., 2017*; *Lichinchi et al., 2016*; *Liu et al., 2023*; *Yang et al., 2022*; *Patil et al., 2016*; *Su et al., 2018*; *Wen et al., 2018*; *Zhou et al., 2015*). Recently, it was uncovered that m6A deposition could, to a considerable extent, be attributed to simple exclusion (100-nt rule), wherein m6A is installed by default at all eligible consensus motifs, unless they are within ~100 nt of a splice junction (*Uzonyi et al., 2023*) in which case their deposition is inhibited by the exon–junction complex (*He et al., 2023*; *Luo et al., 2023*; *Uzonyi et al., 2023*; *Yang et al., 2022*). Here, we propose that changes in m6A levels, across compartments or upon stimuli, may be similarly predictable via a simple model, wherein they may be attributable – at least in part – to changes in mRNA metabolism. Under this model, no changes in active deposition or removal of methylation from specific sites are needed in order to explain changes in m6A levels, which instead merely reflect consequences of m6A-driven mRNA metabolism.

Passive, rather than active, control over m6A levels is somewhat reminiscent of the passive control over DNA methylation in mammalian systems. DNA methylation, like RNA methylation, is controlled by 'writer' enzymes, and can be reversed via 'erasers', hence rendering the 'active' model a viable and attractive one. Nonetheless, considering the dramatic dynamics to which DNA methylation is subject in the germline, these are mediated primarily passively, by dilution of DNA methylation over the course of cellular divisions (*Rougier et al., 1998*). Our findings also connect with recent findings in

neural cells, revealing that differences in relative mRNA localization within cell bodies versus neurites could, to a considerable extent, be attributed to the stability of the mRNA. While mRNA localization in neurons had been primarily thought to be governed actively, via specific RNA binding proteins involved in directing mRNAs to their target locations, this study thus proposed a passive model, wherein the inherent stability of an RNA plays a role in determining its relative localization (*Loedige et al., 2023*).

The tight – yet indirect – interconnectedness of m6A with mRNA metabolic rates (e.g. production and degradation) highlights the need to exercise caution in the interpretation of experiments along these dimensions. In previous studies, it was often hypothesized that changes observed in m6A levels are likely to be predominantly attributed to active recruitment of methylases or demethylases (*Knuckles et al., 2017*; *Su et al., 2018*; *Wei et al., 2018*; *Zhao et al., 2014*; *Zhou et al., 2015*). Our study points at the possibility that such changes may be passive in nature, and attributed to changes in mRNA metabolism, including differences in production, export or decay. Similarly, in previous studies involving perturbation of the m6A pathway were often found to result in diverse changes on the composition (e.g. selective expression of certain splice isoforms) and in the subcellular whereabouts of the transcriptome, based on which the possibility of a direct and causal connection with m6A was hypothesized (*Mao et al., 2019*; *Wang et al., 2015*; *Wen et al., 2018*). Our findings raise the possibility that such changes in m6A methylation levels could, at least in part, be mediated by the redistribution of mRNAs secondary to loss of cytoplasmic m6A-dependent decay. Nonetheless, m6ADyn should only be considered as a framework for testing parsimonious assumptions, and cannot rule out previously proposed mechanisms regarding the impact of m6A on mRNA metabolism or the mechanisms governing m6A dynamics. For instance, we tested whether experimental measurements would be consistent with m6A-facilitated export, and found that under this scenario experimental measurements are not consistent with predictions (*Figure 2—figure supplement 2a–c*). While this may suggest that m6A does not impact export, contrasting a number of studies on the topic, it could also in principle be consistent with a more complex model wherein m6A both impacts both export and mRNA decay (*Edens et al., 2019*; *Hsu et al., 2019*; *Kim et al., 2021*; *Lesbirel et al., 2018*; *Roundtree et al., 2017*). Under such complex scenarios predictions of m6ADyn become a quantitative outcome of the relative impact of m6A on different processing steps, no longer lending themselves to qualitative interpretation. It is also important to highlight that incorporating a general nuclear degradation parameter into m6ADyn appeared to enhance the agreement between the model and some of the observations. In particular, it strengthened the negative correlation between m6A gene levels and total mRNA half-life, suggesting that while the overall outcomes remain largely unchanged, nuclear decay may contribute meaningfully to the dynamics of m6A-mediated regulation. We mostly chose to proceed with the simpler version of the model, as it effectively captures the key dynamics and allows a more intuitive understanding of the model's predictions (*Figure 2—figure supplement 2a–c*).

While the power of m6ADyn lies in its conceptual simplicity, some of its key caveats also lie therein. First, our exploration of m6ADyn is simulation-based: Predictions of m6ADyn are explored using randomly sampled alpha, beta and gamma rates. Being able to experimentally estimate these parameters per gene would allow obtaining quantitative predictions for how m6A levels in each gene would be impacted by changes in production, export and degradation rates. While systematic acquisition of rates capturing different steps in RNA metabolism has been challenging, this is an area marked by recent advances on the basis of metabolic labeling combined with subcellular fractionation (*Müller et al., 2023*; *Smalec et al., 2022*). Second, m6ADyn operates on the assumption of a binary methylation state for genes (methylated vs. unmethylated). This simplification overlooks the likely reality of discrete methylation states for all transcripts of a gene with multiple m6A sites, wherein the susceptibility of a gene to m6A-mediated decay may be a function of its overall m6A levels. Third, m6ADyn may be over-simplistic in many of its assumptions on mRNA metabolism and outcomes of m6A. mRNA metabolism encompasses more than only production, methylation, nuclear export and decay. Nuclear decay, for example, could also play a major role in some cases, although several previous studies have found its role to generally be negligible (*Müller et al., 2023*; *Smalec et al., 2022*). Alternatively, localization with specialized non-membranous subcellular compartments such as P or TIS granules could also impact the susceptibility of transcripts to m6A-mediated decay (*Cougot et al., 2004*; *Sheth and Parker, 2003*). M6ADyn also does not account for nuclear envelope breakdown during cell division, which would redistribute transcripts between nucleus and

cytoplasm. M6ADyn may also be over-simplistic with respect to its assumptions regarding the roles played by m6A. While m6ADyn only considers the well-established role of m6A in controlling cytoplasmic stability, m6A has also been implicated in a range of additional steps on the mRNA life cycle including splicing, nuclear degradation and export. Thus, while the ability of m6ADyn to predict diverse changes in m6A between and across compartments is indicative of its utility, it should not be interpreted as evidence ruling out the existence or relevance of regulatory levels that are not captured as part of m6ADyn.

In line with the above, we wish to highlight several results in which m6ADyn predictions were not fully, or not at all, supported. First, ActD treatment (perturbation of production) and knockdown of the three YTHDF proteins (perturbation of m6A-dependent decay) both resulted in only relatively subtle alterations in m6A levels. Second, there were two predictions pertaining to the relationship between nuclear/cytoplasmic localization and m6A levels for which we were unable to obtain experimental support in the context of production and YTHDF perturbations. (1) Inhibition of production is predicted by m6ADyn to lead to a stronger decrease in m6A levels for cytoplasmic genes than for nuclear counterparts, which we do not observe experimentally (*Figure 3—figure supplement 2a*). (2) Inhibition of m6A-dependent decay is predicted by m6ADyn to lead to a stronger increase in m6A levels in more cytoplasmic genes, in comparison to more nuclear counterparts, which we also do not observe experimentally (*Figure 3—figure supplement 2b*). Third, the quantitative discrepancy between the relatively mild decrease in m6A levels following ActD treatment versus the much more dramatic decrease following CPT treatment is not accounted for by m6ADyn. Finally, also the correlation between degradation rates and the decrease of m6A level was more pronounced following CPT treatment in comparison to ActD treatment (*Figure 3d*, *Figure 3—figure supplement 1c*). These inconsistencies may partially be accounted for by experimental limitations, such as insufficient knockdown of the YTH proteins (or additional factors promoting m6A decay), or complex pleiotropic effects on mRNA metabolism being invoked secondary to the treatment of cells via different lethal inhibitors. Alternatively, these inconsistencies may also reflect oversimplified assumptions of m6ADyn regarding m6A metabolism and the mRNA life cycle.

To conclude, it is important to highlight that changes in m6A levels, both across subcellular compartments and across stimuli, are relatively rare, and typically also subtle. Our findings suggest that many of these changes, when present, may be accounted for by changes in mRNA metabolism, rather than by active control over m6A at specific sites.

## Materials and methods

### Cell line and cell culture

NIH3T3 (mouse), primary MEFs (mouse), MCF7 (human), and HEK293T (human) cells were cultured in Dulbecco's modified Eagle's medium (DMEM) (Biological Industries, 01-052-1A), and the medium was supplemented with 10% fetal bovine serum. The culturing was performed at 37°C and 5% $CO_2$.

### Cytosol-nucleus fractionation protocol

Initially, 2 million cells were collected in cold PBS and centrifuged at $300 \times g$ for 5 min. After centrifugation, the supernatant was discarded, and the cells were resuspended in 150 µl of buffer A (15 mM Tris-Cl pH 8, NaCl at 15 mM, 60 mM KCl, 1 mM EDTA pH 8, 0.5 mM EGTA pH 8) with RNase inhibitor at 10 U/ml added fresh. The cell suspension was mixed with 150 µl of 2× lysis buffer (buffer A with 5% vol/vol NP40 IGEPAL CA-630), and the lysis suspension was incubated for 5 min on ice. The suspension was then centrifuged at $400 \times g$ for 5 min, and 200 µl of the supernatant was carefully transferred to a new tube, which was identified as the cytoplasmic fraction. The residual supernatant from the nuclear pellet was removed and discarded, and the pellet was resuspended in one ml RLN buffer (50 mM Tris-Cl pH 8.0, 140 mM NaCl, 1.5 mM $MgCl_2$, 0.5% NP40 IGEPAL CA-630, 10 mM EDTA pH 8). This step was performed to serve as an additional washing step to remove cytoplasmic leftovers from the intact nucleus pellet. The supernatant was then removed, and the pellet was identified as the nuclear fraction. Both fractions were subsequently used for RNA isolation, performed using TRIzol (Cat 12183-555) according to the manufacturer's instructions.

## Cell treatment assays

The stability experiments were conducted by plating cells on 6 cm plates and allowing them to grow to 80–90% confluency. The cells were treated with 10 µg/ml Actinomycin D (ActD) for the respective timepoints, followed immediately by whole-cell RNA extraction. For the Actinomycin D time course conducted after METTL3 inhibition with STM2457 (*Yankova et al., 2021*), HEK cells were pre-treated with 5 µM inhibitor for 6 hr before Actinomycin D treatment, and the timepoints were taken subsequently. For CPT treatment, 6 µg/ml of the compound was added to cells grown to 80–90% confluency. Control samples were prepared by treating cells with an equivalent volume of DMSO. In both cases, the samples were extracted after 4 hr. For the heat shock time course experiments, cells were grown on 6 cm plates up to 80–90% confluency. The heat shock samples were incubated at 43°C for either 45 or 90 min. Additional samples were incubated at 43°C for 90 min, followed by a recovery incubation at 37°C for either 1 or 4 hr. Control samples were kept at 37°C throughout the experiment. For siRNA-mediated knockdown, cells were plated a day prior and grown to 80–90% confluency. Transfections for siRNA-mediated knockdown were performed following the Dharmafect protocol. For the triple knockdown, the amounts of siRNA and siControl were adjusted accordingly.

RNA isolation was performed using TRIzol (Cat 12183-555) according to the manufacturer's instructions. For library preparation, polyA selection was performed according to the manufacturer's protocol using Dynabeads mRNA Purification Kit (Cat 61006).

For si*Wtap* knockdown (KD), primary MEFs were seeded at a density of 300,000 cells per well in 6-well plates 24 hr prior to transfection with Dharmacon SMARTpool siRNA targeting *Wtap*. Dharmafect 1 was used as the transfection reagent, and the manufacturer's protocol was followed. For each timepoint (control, hs1.5, r1, r4), a final siRNA concentration of 50 nM was prepared in 2 ml of serum-free medium, and 10 µl of Dharmafect 1 was used per sample. Corresponding siCtrl-treated samples (using Dharmacon control siRNA) were prepared for each si*Wtap*-treated sample. After 24 hr of siRNA transfection, the medium was replaced with 2 ml of full DMEM. After an additional 24 hr, a heat shock time-course experiment was performed, and subcellular fractionation was subsequently conducted to isolate nuclear and cytoplasmic fractions. RNA-seq libraries were then prepared from these fractions for further analysis.

## Single-molecule fluorescence in situ hybridization

One day prior to the experiment, coverslips were plated with cells in 6-well plates. The cells were fixed using cold 4% paraformaldehyde for 10 min at room temperature, followed by a rinse with PBS × 1, and permeabilization was performed with 70% cold ethanol for 2 hr (until overnight) at 4°C. On the following day, the samples were washed with SSC × 2 (1 × 150 mM sodium chloride and 15 mM trisodium citrate, adjusted to pH 7.0 with HCl) for 5 min at room temperature, and formamide wash buffers (20% formamide) were prepared. Hybridization buffers (containing 10% vol/vol Dextran Sulfate, 15% formamide, 2 × SSC buffer concentration, 0.02% wt/vol BSA, 2 mM Vanadylribonucleoside, 1 mg/ml *E. coli* tRNA, and nuclease-free water) were mixed with the probes (*Table 1*) at a 1:100 dilution and incubated overnight at 30°C. Post-hybridization washes were conducted using preheated wash buffer (20% formamide) for 30 min at 30°C, and the samples were mounted with a freshly prepared GLOX buffer. DAPI was applied at a 1:8000 dilution for 2 min at room temperature for nuclear counterstaining, and the samples were prepared for microscopy analysis.

### Single-molecule fluorescence in situ hybridization image analysis

Images were taken using a Nikon microscope in the.nd2 stacked format. From the 11 z-stacks, the central five were combined into a single image. The DAPI channel was used to determine the regions of interest (ROIs) for nuclear areas. Cytoplasmic areas were defined as a fixed ellipse surrounding the nucleus, with a vertical and horizontal radius twice the size of the nucleus. Subsequently, the GFP channel was subjected to a Gaussian blur (sigma = 2) and subtracted from the signal channel to remove background autofluorescence. For measuring the poly-dT signal, the signal intensity was

**Table 1.** Single-molecule fluorescence in situ hybridization (smFISH) library probes.

| Probe target | Dye |
| --- | --- |
| polyA | Quasar 570 (Cy 3 Replacement) |

measured, and the mean signal of the respective ROIs (nuclear and cytoplasmic) was compared. For the single-molecule fluorescence in situ hybridization (smFISH) stainings targeting individual genes, the signal was converted into a binary mask using the 'Otsu' thresholding method, and particles within the respective ROIs were measured and compared.

## Site-specific cleavage and radioactive-labeling followed by ligation-assisted extraction and thin-layer chromatography – SCARLET

SCARLET, based on the method described by *Liu et al., 2013*, was performed as outlined by *Garcia-Campos et al., 2019* for three selected m6A sites in genes that showed increased m6A methylation during the cellular heat shock response. The SCARLET assay was conducted on polyA-selected mRNA samples extracted from primary MEFs.

## Transcriptomic analysis and metrics

Each dataset was first trimmed using Cutadapt (*Martin, 2011*). The subsequent paired-end genome alignment for human and mouse mRNA samples was performed against the hg19/mm9 reference genome using STAR v.2.5.3a with default parameters, including the `--alignIntronMax` 1000000 parameter (*Dobin et al., 2013*). Uniquely aligned reads were extracted for further analysis. For gene expression estimation (TPM) or for calculating the expected read counts, rsem-calculate-expression (rsem/1.3.3) was used (*Li and Dewey, 2011*).

## m6A gene index and m6A sample index

m6A-GI and m6A-SI were computed according to *Dierks et al., 2021* using the bam2ReadEnds.R script (*Garcia-Campos et al., 2019*) to extract transcriptome-wide paired-end read coverage and to infer local m6A IP enrichment normalized to the input coverage. For the heat shock time course libraries, TMM normalization of both the Input and IP fractions was performed prior to the generation of m6A-GIs to reduce variability between replicates.

## m6A GLORI gene level

To calculate the m6A gene level, the GLORI sequencing data (GSE210563) from MEFs, as reported by *Yang et al., 2022*, were downloaded. The corresponding fastq files were processed according to *Yang et al., 2022*. Alignment was conducted using HISAT-3N (*Zhang et al., 2021*) with default settings, including the `--base-change` A,G parameter. The resulting BAM files were processed into an.rds file (compatible with RStudio) using txtools (*Garcia-Campos and Schwartz, 2024*), with the mm9 reference genome and a canonical UCSC mm9 gene annotation.

GLORI scores were calculated for annotated adenosines positioned within a DRACH motif environment. The scores were determined by calculating the ratio of unconverted adenosines (A) to the total coverage at each specific site, with filtering criteria set to include only sites with a coverage of at least 20 and a GLORI score greater than 0.1. To calculate the GLORI m6A gene index (GLORI-GI), the sum of GLORI scores from all eligible DRACH motifs within each gene was used and normalized by the total number of adenosines within the respective gene.

## Nuclear:cytoplasmic relative localization

RSEM (*Li and Dewey, 2011*) was used to determine the expected read counts for the nuclear and cytoplasmic fractions. Cross-sample normalization for each of the two subcellular fractions was performed using the TMM function from the NOISeq package in R (*Tarazona et al., 2011*). Following normalization, log fold-changes between the normalized nuclear and cytoplasmic counts were calculated.

## mRNA half-life estimation

mRNA decay rate estimation was performed through linear modeling using the lm() function from R 4.0.1 stats (version 3.6.2) on RSEM-based TPM estimates for all timepoints of the Actinomycin D time course. The estimated slope from the linear model was used as the decay rate (k). Only negative slopes with a p-value <0.05 were considered for further analysis. The corresponding half-lives were calculated based on these decay rates.

$$HL = \frac{ln\,(2)}{-k}.$$

## PCA analysis

The PCA was conducted using the prcomp() function from R 4.0.1 stats (version 3.6.2) on a matrix of log2-transformed, scaled m6A-GIs for each sample. Only genes for which an m6A-GI could be estimated at every timepoint (sample and replicate) were included in the analysis.

## 'm6ADyn' model calculation

m6ADyn is a differential equation-based model for simulating mRNA metabolism. In m6ADyn, transcripts are produced in the nucleus at a production rate ($\alpha$), exported from the nucleus to the cytoplasm at an export rate ($\beta$), and degraded in the cytoplasm at a decay rate ($\gamma$). Separate sets of rates are defined for non-m6A-harboring transcripts ($\alpha^A$, $\beta^A$, and $\gamma^A$) and m6A-harboring transcripts ($\alpha^{m6A}$, $\beta^{m6A}$, and $\gamma^{m6A}$). Unless specified otherwise, in our implementations and simulations, the decay rate for m6A-harboring transcripts is assumed to be higher than that of non-m6A-harboring transcripts ($\gamma^{m6A} > \gamma^A$), while the export rates for both transcript types are assumed to be equal ($\beta^A = \beta^{m6A}$).

The model is defined by the following two differential equations:

$$(I)\ \frac{dNuc}{dt} = \alpha - \beta \cdot Nuc$$

$$(II)\ \frac{dCyt}{dt} = \beta \cdot Nuc - \gamma \cdot Cyt$$

Whereby the nuclear (Nuc) and cytoplasmic (Cyt) levels refer to the transcript levels of a gene within the nucleus and cytoplasm, respectively. The nuclear and cytoplasmic transcript levels are derived under steady-state conditions, where no change in transcript concentration occurs over time ($\frac{dNuc}{dt} = \frac{dCyt}{dt} = 0$).

$$(I)\ 0 = \alpha - \beta \cdot Nuc$$

$$Nuc = \frac{\alpha}{\beta}$$

$$(II)\ 0 = \beta \cdot Nuc - \gamma \cdot Cyt$$

$$Cyt = \frac{\alpha}{\gamma}$$

The availability of nuclear and cytoplasmic abundances of both m6A-containing and m6A-lacking transcript counterparts allows the following values to be derived under steady-state conditions:

$$Gene\ level = Cyt^{m6A} + Nuc^{m6A} + Cyt^A + Nuc^A \tag{1}$$

$$Nuc\ m6A\ level = \frac{Nuc^{m6A}}{Nuc^A + Nuc^{m6A}} \tag{2}$$

$$Cyt\ m6A\ level = \frac{Cyt^{m6A}}{Cyt^A + Cyt^{m6A}} \tag{3}$$

$$m6A\ level = \frac{Cyt^{m6A} + Nuc^{m6A}}{Cyt^A + Nuc^A + Cyt^{m6A} + Nuc^{m6A}} \tag{4}$$

$$Nuc:Cyt\ ratio = \frac{Nuc^A + Nuc^{m6A}}{Cyt^A + Cyt^{m6A}} \tag{5}$$

For the derivation of nuclear and cytoplasmic transcript abundances outside of steady-state conditions, the R package 'MOSAIC' (*Pruim et al., 2017*) was used to simulate changes in transcript levels over time. This package was also employed to simulate stability rates by fitting a linear model to m6ADyn-derived log-transformed mRNA levels along a time course following the inhibition of transcription ($\alpha^A = \alpha^{m6A} = 0$), thereby mimicking an experiment based on transcriptional inhibition.

For the simulation of genes, the required rates ($\alpha^A$, $\alpha^{m6A}$, $\beta$, and $\gamma$) were sampled independently. The rates were drawn from a gamma distribution, which, by default, was set with a rate parameter of 1

**Table 2.** List of the all the NGS datasets used in this manuscript.

| Dataset | Source | Figure |
| --- | --- | --- |
| Nuc:Cyt fractionation experiment followed by m6A-seq2, NIH3T3 | GSE272629 | *Figure 2* |
| Nuc:Cyt fractionation followed by RNA-seq, WT, and Mettl3 KO mESC | GSE272629 | *Figure 2* |
| m6A-seq, WT, and Mettl3 KO mESC | *Garcia-Campos et al., 2019* | *Figure 2* |
| Actinomycin D time course followed by RNA-seq mESC WT and Mettl3 KO | *Ke et al., 2017* | *Figure 2* |
| Actinomycin D time course followed by RNA-seq WT and METTL3 Inhibitor, HEK | GSE272629 | *Figure 2—figure supplement 1* |
| m6A-seq experiment, HEK293 | *Garcia-Campos et al., 2019* | *Figure 2—figure supplement 1* |
| Fractionation followed by RNA-seq HepG2 | GSE30567 (ENCODE) | *Figure 2—figure supplement 1* |
| Nuc:Cyt fractionation experiment followed by m6A-seq2, MEFs | GSE272629 | *Figure 2—figure supplement 1* |
| GLORI, MEF NHS | *Liu et al., 2023* | *Figure 2—figure supplement 1* |
| CPT treatement followed by m6A-seq2, MCF7 | GSE272629 | *Figure 3* |
| Actinomycin D time course followed by m6A-seq2, A9 cells | GSE272629 | *Figure 3* |
| m6A-seq2 of YTHDF1-3 knockdown, NIH3T3 | GSE272629 | *Figure 3* |
| Nuc:Cyt fractionation YTHDF1-3 knockdown 3T3 cells | GSE272629 | *Figure 3* |
| Heat stress followed by m6A-seq2, WT pMEF | GSE272629 | *Figure 4* |
| Heat stress followed by m6A-seq2, HSF1KO pMEF | GSE272629 | *Figure 4* |
| Nuc:Cyt fractionation experiment followed by RNA-seq after heat stress, WT pMEF, and *Wtap* KD | GSE272629 | *Figure 4* |
| Nuc:Cyt fractionation experiment followed by RNA-seq after heat stress, HSF1-KO pMEF | GSE272629 | *Figure 4* |
| PRO-seq after heat stress, MEF | *Mahat et al., 2016* | *Figure 4* |

and a shape parameter of 1, unless specified otherwise. Unless stated otherwise, export rates (β) were assumed to be equal for both methylated and unmethylated transcripts ($\beta^A = \beta^{m6A}$), while degradation rates for methylated transcripts were set to be higher by a factor (S) of ten ($\gamma^{m6A} = 10 * \gamma^A$). An example of such a simulation was provided in R code and included in *Supplementary file 1*.

## m6ADyn with nuclear degradation

To simulate the effects of nuclear degradation, nuclear decay (δ) was introduced as an additional rate parameter. The inclusion of this parameter resulted in minor modifications to the equations, accounting for the degradation of transcripts within the nucleus alongside production, export, and cytoplasmic decay rates.

$$(I.I) \quad \frac{dNuc}{dt} = \alpha - (\beta \cdot Nuc) - (\delta \cdot Nuc) \quad (II.I) \quad \frac{dCyt}{dt} = \beta \cdot Nuc - \gamma \cdot Cyt$$

Solving the equations for steady-state conditions ($\frac{dN}{dt} = \frac{dC}{dt} = 0$).

$$(I.I) \quad 0 = \alpha - \beta \cdot Nuc$$

$$Nuc = \frac{\alpha}{\beta + \delta}$$

## Data availability

All the NGS datasets generated during the course of this study have been deposited in GSE272629 (*Table 2*).

## Code availability

A demonstration and associated source code for m6ADyn simulation is added in *Supplementary file 1*. The used in-house python script for demultiplexing of multiplexed RNA-seq samples as well as the scripts bam2ReadEnds.R and txtools *Garcia-Campos and Schwartz, 2024* used for paired-end transcript coverage are stored in https://github.com/SchwartzLab/bam2ReadEnds (copy archived at *Dierks, 2025*).

## Acknowledgements

SS is funded by the Israel Science Foundation (grant no. 913/21), the European Research Council (ERC) under the European Union's Horizon 2020 research and innovation programme (grant no. 101000970). RSS is supported by the Israel Science Foundation (grant number 395-21) and the European Research Council (ERC grant number 101043300). RSS is the incumbent of the Robert and Yadelle Sklare Professorial Chair in Biochemistry.

## Additional information

### Funding

| Funder | Grant reference number | Author |
| --- | --- | --- |
| Israel Science Foundation | 913/21 | Schraga Schwartz |
| HORIZON EUROPE European Research Council | 10.3030/101000970 | Schraga Schwartz |
| European Research Council | 10.3030/101043300 | Ruth Scherz-Shouval |
| Israel Science Foundation | 395-21 | Ruth Scherz-Shouval |
| Robert and Yadelle Sklare Professorial Chair in Biochemistry | | Ruth Scherz-Shouval |

The funders had no role in study design, data collection, and interpretation, or the decision to submit the work for publication.

### Author contributions

David Dierks, Conceptualization, Data curation, Software, Formal analysis, Validation, Visualization, Methodology, Writing – original draft, Project administration, Writing – review and editing; Ran Shachar, Ronit Nir, Anna Uzonyi, Ursula Toth, Walter Rossmanith, Lior Lasman, Data curation; Miguel Angel Garcia-Campos, Software; David Wiener, Formal analysis; Boris Slobodin, Resources, Data curation; Jacob H Hanna, Resources, Supervision; Yaron Antebi, Formal analysis, Methodology; Ruth Scherz-Shouval, Conceptualization, Resources, Data curation, Supervision, Funding acquisition, Validation, Investigation, Visualization, Methodology, Writing – original draft, Project administration, Writing – review and editing; Schraga Schwartz, Conceptualization, Resources, Data curation, Software, Formal analysis, Supervision, Funding acquisition, Validation, Investigation, Visualization, Methodology, Writing – original draft, Project administration, Writing – review and editing

### Author ORCIDs

David Dierks ⓘ https://orcid.org/0000-0003-1515-7222
David Wiener ⓘ https://orcid.org/0000-0002-7435-4253
Yaron Antebi ⓘ https://orcid.org/0000-0002-5771-6814
Ruth Scherz-Shouval ⓘ https://orcid.org/0000-0002-4570-121X
Schraga Schwartz ⓘ https://orcid.org/0000-0002-3671-9709

Reviewer #1 (Public review): https://doi.org/10.7554/eLife.100448.3.sa1
Reviewer #2 (Public review): https://doi.org/10.7554/eLife.100448.3.sa2
Reviewer #3 (Public review): https://doi.org/10.7554/eLife.100448.3.sa3
Author response https://doi.org/10.7554/eLife.100448.3.sa4

## Additional files

### Supplementary files
MDAR checklist

Supplementary file 1. R Code for Exemplary m6ADyn Model Analysis.

### Data availability
Sequencing data have been deposited in GEO under the accession GSE272629.

The following dataset was generated:

| Author(s) | Year | Dataset title | Dataset URL | Database and Identifier |
|---|---|---|---|---|
| Dierks D, Schwartz S, Shachar R, Nir R, Garcia-Campos MA, Uzonyi A, Toth U, Rossmanith W, Lasman L, Slobodin B, Hanna JH, Antebi Y, Scherz-Shouval R, Wiener D | 2024 | Passive shaping of intra- and intercellular m6A dynamics via mRNA metabolism | https://www.ncbi.nlm.nih.gov/geo/query/acc.cgi?acc=GSE272629 | NCBI Gene Expression Omnibus, GSE272629 |

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
