## [Editor Report · eLife Assessment]

This study presents a **fundamental** finding on how levels of m6A levels are controlled, invoking a consolidated model where degradation of modified RNAs in the cytoplasm plays a primary role in shaping m6A patterns and dynamics, rather than needing active regulation by m6A erasers and other related processes. The evidence is **compelling** through its use of transcriptome-wide data and mechanistic modeling. Relevant for any reader with an interest in RNA metabolism, this new framework consolidates previous observations and highlights the importance of careful experimentation for evaluation m6A levels.

---

## [Referee Report · Reviewer #1 (Public review)]

Here, the authors propose that changes in m6A levels may be predictable via a simple model that is based exclusively on mRNA metabolic events. Under this model, m6A mRNAs are "passive" victims of RNA metabolic events with no "active" regulatory events needed to modulate their levels by m6A writers, readers, or erasers; looking at changes in RNA transcription, RNA export, and RNA degradation dynamics is enough to explain how m6A levels change over time.

The relevance of this study is extremely high at this stage of the epitranscriptome field. This compelling paper is in line with more and more recent studies showing how m6A is a constitutive mark reflecting overall RNA redistribution events. At the same time, it reminds every reader to carefully evaluate changes in m6A levels if observed in their experimental setup. It highlights the importance of performing extensive evaluations on how much RNA metabolic events could explain an observed m6A change.

---

## [Referee Report · Reviewer #2 (Public review)]

Dierks et al. investigate the impact of m6A RNA modifications on the mRNA life cycle, exploring the links between transcription, cytoplasmic RNA degradation and subcellular RNA localization. Using transcriptome-wide data and mechanistic modelling of RNA metabolism, the authors demonstrate that a simplified model of m6A primarily affecting cytoplasmic RNA stability is sufficient to explain the nuclear-cytoplasmic distribution of methylated RNAs and the dynamic changes in m6A levels upon perturbation. Based on multiple lines of evidence, they propose that passive mechanisms based on the restricted decay of methylated transcripts in the cytoplasm play a primary role in shaping condition-specific m6A patterns and m6A dynamics. The authors support their hypothesis with multiple large-scale datasets and targeted perturbation experiments. Overall, the authors present compelling evidence for their model which has the potential to explain and consolidate previous observations on different m6A functions, including m6A-mediated RNA export.

---

## [Referee Report · Reviewer #3 (Public review)]

Summary:

This manuscript works with a hypothesis where the overall m6A methylation levels in cells is influenced by mRNA metabolism (sub-cellular localization and decay). The basic assumption is that m6A causes Mrna decay and this happens in the cytoplasm. They go on to experimentally test their model to confirm its predictions. This is confirmed by sub-cellular fractionation experiments which shows high m6A levels in the nuclear RNA. Nuclear localized RNAs have higher methylation. Using a heat shock model, they demonstrate that RNAs with increased nuclear localization or transcription, are methylated at higher levels. Their overall argument is that changes in m6A levels is rather determined by passive processes that are influenced by RNA processing/metabolism. However, it should be considered that erasers have their roles under specific environments (early embryos or germline) and are not modelled by the cell culture systems used here.

Strengths:

This is a thought-provoking series of experiments that challenge the idea that active mechanisms of recruitment or erasure are major determinants for m6A distribution and levels.

Comments on revisions:

The authors have done a good job with the revision.

---

## [Author Response]

The following is the authors’ response to the original reviews.

**Public Reviews:**

**Reviewer #1 (Public review):**
Summary:Here, the authors propose that changes in m6A levels may be predictable via a simple model that is based exclusively on mRNA metabolic events. Under this model, m6A mRNAs are "passive" victims of RNA metabolic events with no "active" regulatory events needed to modulate their levels by m6A writers, readers, or erasers; looking at changes in RNA transcription, RNA export, and RNA degradation dynamics is enough to explain how m6A levels change over time.The relevance of this study is extremely high at this stage of the epi transcriptome field. This compelling paper is in line with more and more recent studies showing how m6A is a constitutive mark reflecting overall RNA redistribution events. At the same time, it reminds every reader to carefully evaluate changes in m6A levels if observed in their experimental setup. It highlights the importance of performing extensive evaluations on how much RNA metabolic events could explain an observed m6A change.Weaknesses:It is essential to notice that m6ADyn does not exactly recapitulate the observed m6A changes. First, this can be due to m6ADyn's limitations. The authors do a great job in the Discussion highlighting these limitations. Indeed, they mention how m6ADyn cannot interpret m6A's implications on nuclear degradation or splicing and cannot model more complex scenario predictions (i.e., a scenario in which m6A both impacts export and degradation) or the contribution of single sites within a gene.Secondly, since predictions do not exactly recapitulate the observed m6A changes, "active" regulatory events may still play a partial role in regulating m6A changes. The authors themselves highlight situations in which data do not support m6ADyn predictions. Active mechanisms to control m6A degradation levels or mRNA export levels could exist and may still play an essential role.

We are grateful for the reviewer’s appreciation of our findings and their implications, and are in full agreement with the reviewer regarding the limitations of our model, and the discrepancies in some cases - with our experimental measurements, potentially pointing at more complex biology than is captured by m6ADyn. We certainly cannot dismiss the possibility that active mechanisms may play a role in shaping m6A dynamics at some sites, or in some contexts. Our study aims to broaden the discussion in the field, and to introduce the possibility that passive models can explain a substantial extent of the variability observed in m6A levels.

(1) "We next sought to assess whether alternative models could readily predict the positive correlation between m6A and nuclear localization and the negative correlations between m6A and mRNA stability. We assessed how nuclear decay might impact these associations by introducing nuclear decay as an additional rate, δ. We found that both associations were robust to this additional rate (Supplementary Figure 2a-c)."Based on the data, I would say that model 2 (m6A-dep + nuclear degradation) is better than model 1. The discussion of these findings in the Discussion could help clarify how to interpret this prediction. Is nuclear degradation playing a significant role, more than expected by previous studies?

This is an important point, which we’ve now clarified in the discussion. Including nonspecific nuclear degradation in the m6ADyn framework provides a model that better aligns with the observed data, particularly by mitigating unrealistic predictions such as excessive nuclear accumulation for genes with very low sampled export rates. This adjustment addresses potential artifacts in nuclear abundance and half-life estimations. However, we continued to use the simpler version of m6ADyn for most analyses, as it captures the key dynamics and relationships effectively without introducing additional complexity. While including nuclear degradation enhances the model's robustness, it does not fundamentally alter the primary conclusions or outcomes. This balance allows for a more straightforward interpretation of the results.

(2) The authors classify m6A levels as "low" or "high," and it is unclear how "low" differs from unmethylated mRNAs.

We thank the reviewer for this observation. We analyzed gene methylation levels using the m6A-GI (m6A gene index) metric, which reflects the enrichment of the IP fraction across the entire gene body (CDS + 3UTR). While some genes may have minimal or no methylation, most genes likely exist along a spectrum from low to high methylation levels. Unlike earlier analyses that relied on arbitrary thresholds to classify sites as methylated, GLORI data highlight the presence of many low-stoichiometry sites that are typically overlooked. To capture this spectrum, we binned genes into equal-sized groups based on their m6A-GI values, allowing a more nuanced interpretation of methylation patterns as a continuum rather than a binary or discrete classification (e.g. no- , low- , high methylation).

(3) The authors explore whether m6A changes could be linked with differences in mRNA subcellular localization. They tested this hypothesis by looking at mRNA changes during heat stress, a complex scenario to predict with m6ADyn. According to the collected data, heat shock is not associated with dramatic changes in m6A levels. However, the authors observe a redistribution of m6A mRNAs during the treatment and recovery time, with highly methylated mRNAs getting retained in the nucleus being associated with a shorter half-life, and being transcriptional induced by HSF1. Based on this observation, the authors use m6Adyn to predict the contribution of RNA export, RNA degradation, and RNA transcription to the observed m6A changes. However:(a) Do the authors have a comparison of m6ADyn predictions based on the assumption that RNA export and RNA transcription may change at the same time?

We thank the reviewer for this point. Under the simple framework of m6ADyn in which RNA transcription and RNA export are independent of each other, the effect of simultaneously modulating two rates is additive. In Author response image 1, we simulate some scenarios wherein we simultaneously modulate two rates. For example, transcriptional upregulation and decreased export during heat shock could reinforce m6A increases, whereas transcriptional downregulation might counteract the effects of reduced export. Note that while production and export can act in similar or opposing directions, the former can only lead to temporary changes in m6A levels but without impacting steady-state levels, whereas the latter (changes in export) can alter steady-state levels. We have clarified this in the manuscript results to better contextualize how these dynamics interact.

**Author response image 1. sa4fig1:** m6ADyn predictions of m6A gene levels (left) and Nuc to Cyt ratio (right) upon varying perturbations of a sampled gene. The left panel depicts the simulated dynamics of log2-transformed m6A gene levels under varying conditions. The lines represent the following perturbations: (1) export is reduced to 10% (β), (2) production is increased 10-fold (α) while export is reduced to 10% (β), (3) export is reduced to 10% (β) and production is reduced to 10% (α), and (4) export is only decreased for methylated transcripts (β^m6A) to 10%. The right panel shows the corresponding nuclear:cytoplasmic (log2 Nuc:Cyt) ratios for perturbations 1 and 4.

(b) They arbitrarily set the global reduction of export to 10%, but I'm not sure we can completely rule out whether m6A mRNAs have an export rate during heat shock similar to the non-methylated mRNAs. What happens if the authors simulate that the block in export could be preferential for m6A mRNAs only?

We thank the reviewer for this interesting suggestion. While we cannot fully rule out such a scenario, we can identify arguments against it being an exclusive explanation. Specifically, an exclusive reduction in the export rate of methylated transcripts would be expected to increase the relationship between steady-state m6A levels (the ratio of methylated to unmethylated transcripts) and changes in localization, such that genes with higher m6A levels would exhibit a greater relative increase in the nuclear-to-cytoplasmic (Nuc:Cyt) ratio. However, the attached analysis shows only a weak association during heat stress, where genes with higher m6A-GI levels tend to increase just a little more in the Nuc:Cyt ratio, likely due to cytoplasmic depletion. A global reduction of export (β 10%) produces a similar association, while a scenario where only the export of methylated transcripts is reduced (β^m6A 10%) results in a significantly stronger association (Author response image 2). This supports the plausibility of a global export reduction. Additionally, genes with very low methylation levels in control conditions also show a significant increase in the Nuc:Cyt ratio, which is inconsistent with a scenario of preferential export reduction for methylated transcripts (data not shown).

**Author response image 2. sa4fig2:** Wild-type MEFs m6A-GIs (x-axis) vs.fold change nuclear:cytoplasmic localization heat shock 1.5 h and control (y-axis), Pearson’s correlation indicated (left panel). m6ADyn, rates sampled for 100 genes based on gamma distributions and simulation based on reducing the global export rate (β) to 10% (middle panel). m6ADyn simulation for reducing the export rate for m6A methylated transcripts (β^m6A) to 10% (right panel).

(c) The dramatic increase in the nucleus: cytoplasmic ratio of mRNA upon heat stress may not reflect the overall m6A mRNA distribution upon heat stress. It would be interesting to repeat the same experiment in METTL3 KO cells. Of note, m6A mRNA granules have been observed within 30 minutes of heat shock. Thus, some m6A mRNAs may still be preferentially enriched in these granules for storage rather than being directly degraded. Overall, it would be interesting to understand the authors' position relative to previous studies of m6A during heat stress.

The reviewer suggests that methylation is actively driving localization during heat shock, rather than being passively regulated. To address this question, we have now knocked down WTAP, an essential component of the methylation machinery, and monitored nuclear:cytoplasmic localization over the course of a heat shock response. Even with reduced m6A levels, high PC1 genes exhibit increased nuclear abundance during heat shock. Notably, the dynamics of this trend are altered, with the peak effect delayed from 1.5h heat shock in siCTRL samples to 4 hours in siWTAP samples (Supplementary Figure 4). This finding underscores that m6A is not the primary driver of these mRNA localization changes but rather reflects broader mRNA metabolic shifts during heat shock. These findings have been added as a panel e to Supplementary Figure 4.

(d) Gene Ontology analysis based on the top 1000 PC1 genes shows an enrichment of GOs involved in post-translational protein modification more than GOs involved in cellular response to stress, which is highlighted by the authors and used as justification to study RNA transcriptional events upon heat shock. How do the authors think that GOs involved in post-translational protein modification may contribute to the observed data?

High PC1 genes exhibit increased methylation and a shift in nuclear-to-cytoplasmic localization during heat stress. While the enriched GO terms for these genes are not exclusively related to stress-response proteins, one could speculate that their nuclear retention reduces translation during heat stress. The heat stress response genes are of particular interest, which are massively transcriptionally induced and display increased methylation. This observation supports m6ADyn predictions that elevated methylation levels in these genes are driven by transcriptional induction rather than solely by decreased export rates.

(e) Additionally, the authors first mention that there is no dramatic change in m6A levels upon heat shock, "subtle quantitative differences were apparent," but then mention a "systematic increase in m6A levels observed in heat stress". It is unclear to which systematic increase they are referring to. Are the authors referring to previous studies? It is confusing in the field what exactly is going on after heat stress. For instance, in some papers, a preferential increase of 5'UTR m6A has been proposed rather than a systematic and general increase.

We thank the reviewer for raising this point. In our manuscript, we sought to emphasize, on the one hand, the fact that m6A profiles are - at first approximation - “constitutive”, as indicated by high Pearson correlations between conditions (Supplementary Figure 4a). On the other hand, we sought to emphasize that the above notwithstanding, subtle quantitative differences are apparent in heat shock, encompassing large numbers of genes, and these differences are coherent with time following heat shock (and in this sense ‘systematic’), rather than randomly fluctuating across time points. Based on our analysis, these changes do not appear to be preferentially enriched at 5′UTR sites but occur more broadly across gene bodies (potentially a slight 3’ bias). A quick analysis of the HSF1-induced heat stress response genes, focusing on their relative enrichment of methylation upon heat shock, shows that the 5'UTR regions exhibit a roughly similar increase in methylation after 1.5 hours of heat stress compared to the rest of the gene body (Author response image 3). A prominent previous publication (Zhou et al. 2015) suggested that m6A levels specifically increase in the 5'UTR of HSPA1A in a YTHDF2- and HSF1-dependent manner, and highlighted the role of 5'UTR m6A methylation in regulating cap-independent translation, our findings do not support a 5'UTR-specific enrichment. However, we do observe that the methylation changes are still HSF1-dependent. Off note, the m6A-GI (m6A gene level) as a metric that captures the m6A enrichment of gene body excluding the 5’UTR, due to an overlap of transcription start site associated m6Am derived signal.

**Author response image 3. sa4fig3:** Fold change of m6A enrichment (m6A-IP / input) comparing 1. 5 h heat shock and control conditions for 5UTR region and the rest of the gene body (CDS and 3UTR) in the 10 HSF! dependent stress response genes.

**Reviewer #2 (Public review):**
Dierks et al. investigate the impact of m6A RNA modifications on the mRNA life cycle, exploring the links between transcription, cytoplasmic RNA degradation, and subcellular RNA localization. Using transcriptome-wide data and mechanistic modelling of RNA metabolism, the authors demonstrate that a simplified model of m6A primarily affecting cytoplasmic RNA stability is sufficient to explain the nuclear-cytoplasmic distribution of methylated RNAs and the dynamic changes in m6A levels upon perturbation. Based on multiple lines of evidence, they propose that passive mechanisms based on the restricted decay of methylated transcripts in the cytoplasm play a primary role in shaping condition-specific m6A patterns and m6A dynamics. The authors support their hypothesis with multiple large-scale datasets and targeted perturbation experiments. Overall, the authors present compelling evidence for their model which has the potential to explain and consolidate previous observations on different m6A functions, including m6A-mediated RNA export.

We thank the reviewer for the spot-on suggestions and comments on this manuscript.

**Reviewer #3 (Public review):**
Summary:This manuscript works with a hypothesis where the overall m6A methylation levels in cells are influenced by mRNA metabolism (sub-cellular localization and decay). The basic assumption is that m6A causes mRNA decay and this happens in the cytoplasm. They go on to experimentally test their model to confirm its predictions. This is confirmed by sub-cellular fractionation experiments which show high m6A levels in the nuclear RNA. Nuclear localized RNAs have higher methylation. Using a heat shock model, they demonstrate that RNAs with increased nuclear localization or transcription, are methylated at higher levels. Their overall argument is that changes in m6A levels are rather determined by passive processes that are influenced by RNA processing/metabolism. However, it should be considered that erasers have their roles under specific environments (early embryos or germline) and are not modelled by the cell culture systems used here.Strengths:This is a thought-provoking series of experiments that challenge the idea that active mechanisms of recruitment or erasure are major determinants for m6A distribution and levels.

We sincerely thank the reviewer for their thoughtful evaluation and constructive feedback.

**Recommendations for the authors:**

**Reviewer #1 (Recommendations for the authors):**
(1) Supplementary Figure 5A Data: Please double-check the label of the y-axis and the matching legend.

We corrected this.

(2) A better description of how the nuclear: cytoplasmic fractionation is performed.

We added missing information to the Material & Methods section.

(3) Rec 1hr or Rec 4hr instead of r1 and r4 to indicate the recovery.

For brevity in Figure panels, we have chosen to stick with r1 and r4.

(4) Figure 2D: are hours plotted?

Plotted is the fold change (FC) of the calculated half-lives in hours (right). For the model (left) hours are the fold change of a dimension-less time-unit of the conditions with m6A facilitated degradation vs without. We have now clarified this in the legend.

(5) How many genes do we have in each category? How many genes are you investigating each time?

We thank the reviewer for this question. In all cases where we binned genes, we used equal-sized bins of genes that met the required coverage thresholds. We have reviewed the manuscript to ensure that the number of genes included in each analysis or the specific coverage thresholds used are clearly stated throughout the text.

(6) Simulations on 1000 genes or 2000 genes?

We thank the reviewer for this question and went over the text to correct for cases in which this was not clearly stated.

**Reviewer #2 (Recommendations for the authors):**
Specific comments:(1) The manuscript is very clear and well-written. However, some arguments are a bit difficult to understand. It would be helpful to clearly discriminate between active and passive events. For example, in the sentence: "For example, increasing the m6A deposition rate (⍺m6A) results in increased nuclear localization of a transcript, due to the increased cytoplasmic decay to which m6A-containing transcripts are subjected", I would directly write "increased relative nuclear localization" or "apparent increase in nuclear localization".

We thank the reviewer for this careful observation. We have modified the quoted sentence, and also sought to correct additional instances of ambiguity in the text.

Also, it is important to ensure that all relationships are described correctly. For example, in the sentence: "This model recovers the positive association between m6A and nuclear localization but gives rise to a positive association between m6A and decay", I think "decay" should be replaced with "stability". Similarly, the sentence: "Both the decrease in mRNA production rates and the reduction in export are predicted by m6ADyn to result in increasing m6A levels, ..." should it be "Both the increase in mRNA production and..."?

We have corrected this.

This sentence was difficult for me to understand: "Our findings raise the possibility that such changes could, at least in part, also be indirect and be mediated by the redistribution of mRNAs secondary to loss of cytoplasmic m6A-dependent decay." Please consider rephrasing it.

We rephrased this sentence as suggested.

(2) Figure 2d: "A final set of predictions of m6ADyn concerns m6A-dependent decay. m6ADyn predicts that (a) cytoplasmic genes will be more susceptible to increased m6A mediated decay, independent of their m6A levels, and (b) more methylated genes will undergo increased decay, independently of their relative localization (Figure 2d left) ... Strikingly, the experimental data supported the dual, independent impact of m6A levels and localization on mRNA stability (Figure 2d, right)."I do not understand, either from the text or from the figure, why the authors claim that m6A levels and localization independently affect mRNA stability. It is clear that "cytoplasmic genes will be more susceptible to increased m6A mediated decay", as they always show shorter half-lives (top-to-bottom perspective in Figure 2d). Nonetheless, as I understand it, the effect is not "independent of their m6A levels", as half-lives are clearly the shortest with the highest m6A levels (left-to-right perspective in each row).

The two-dimensional heatmaps allow for exploring conditional independence between conditions. If an effect (in this case delta half-life) is a function of the X axis (in this case m6A levels), continuous increases should be seen going from one column to another. Conversely, if it is a function of the Y axis (in this case localization), a continuous effect should be observed from one row to another. Given that effects are generally observed both across rows and across columns, we concluded that the two act independently. The fact that half-life is shortest when genes are most cytoplasmic and have the highest m6A levels is therefore not necessarily inconsistent with two effects acting independently, but instead interpreted by us as the additive outcome of two independent effects. Having said this, a close inspection of this plot does reveal a very low impact of localization in contexts where m6A levels are very low, which could point at some degree of synergism between m6A levels and localization. We have therefore now revised the text to avoid describing the effects as "independent."

(3) The methods part should be extended. For example, the description of the mRNA half-life estimation is far too short and lacks details. Also, information on the PCA analysis (Figure 4e & f) is completely missing. The code should be made available, at least for the differential model.

We thank the reviewer for this point and expanded the methods section on mRNA stability analysis and PCA. Additionally, we added a supplementary file, providing R code for a basic m6ADyn simulation of m6A depleted to normal conditions (added Source Code 1).

(4) Figure 4e, f: The authors use a PCA analysis to achieve an unbiased ranking of genes based on their m6A level changes. From the present text and figures, it is unclear how this PCA was performed. Besides a description in the methods sections, the authors could show additional evidence that the PCA results in a meaningful clustering and that PC1 indeed captures induced/reduced m6A level changes for high/low-PC1 genes.

We have added passages to the text, hoping to clarify the analysis approach.

(5) In Figure 4i, I was surprised about the m6A dynamics for the HSF1-independent genes, with two clusters of increasing or decreasing m6A levels across the time course. Can the model explain these changes? Since expression does not seem to be systematically altered, are there differences in subcellular localization between the two clusters after heat shock?

A general aspect of our manuscript is attributing changes in m6A levels during heat stress to alterations in mRNA metabolism, such as production or export. As shown in Supplementary Figure 4d, even in WT conditions, m6A level changes are not strictly associated with apparent changes in expression, but we try to show that these are a reflection of the decreased export rate. In the specific context of HSF1-dependent stress response genes, we observe a clear co-occurrence of increased m6A levels with increased expression levels, which we propose to be attributed to enhanced production rates during heat stress. This suggests that transcriptional induction can drive the apparent rise in m6A levels. We try to control this with the HSF1 KO cells, in which the m6A level changes, as the increased production rates are absent for the specific cluster of stress-induced genes, further supporting the role of transcriptional activation in shaping m6A levels for these genes. For HSF1-independent genes, the HSF-KO cells mirror the behavior of WT conditions when looking at 500 highest and lowest PC1 (based on the prior analysis in WT cells), suggesting that changes in m6A levels are primarily driven by altered export rates rather than changes in production.

Among the HSF1 targets, Hspa1a seems to show an inverse behaviour, with the highest methylation in ctrl, even though expression strongly goes up after heat shock. Is this related to the subcellular localization of this particular transcript before and after heat shock?

Upon reviewing the heat stress target genes, we identified an issue with the proper labeling of the gene symbols, which has now been corrected (Figure 4 panel i). The inverse behavior observed for Hspb1 and partially for Hsp90aa1 is not accounted for by the m6ADyn model, and is indeed an interesting exception with respect to all other induced genes. Further investigation will be required to understand the methylation dynamics of Hspb1 during the response to heat stress.

**Reviewer #3 (Recommendations for the authors):**
Page 4. Indicate reference for "a more recent study finding reduced m6A levels in chromatin-associated RNA.".

We thank the reviewer for this point and added two publications with a very recent one, both showing that chromatin-associated nascent RNA has less m6A methylation

The manuscript is perhaps a bit too long. It took me a long time to get to the end. The findings can be clearly presented in a more concise manner and that will ensure that anyone starting to read will finish it. This is not a weakness, but a hope that the authors can reduce the text.

We have respectfully chosen to maintain the length of the manuscript. The model, its predictions and their relationship to experimental observations are somewhat complex, and we felt that further reduction of the text would come at the expense of clarity.